# Hydro-geomorphological modelling of leaky wooden dam efficacy from reach to catchment scale with CAESAR-Lisflood 1.9j

Joshua M. Wolstenholme[1,2], Christopher J. Skinner[3,2], David Milan[4], Robert E. Thomas[2] and Daniel R. Parsons[1]

[1]Geography and Environment, Loughborough University, Loughborough, UK
[2]Energy and Environment Institute, University of Hull, Hull, UK
[3]FloodSkinner, UK
[4]School of Environmental Sciences, University of Hull, Hull, UK

*Correspondence to*: Joshua M. Wolstenholme (j.m.wolstenholme@lboro.ac.uk)

**Abstract**. Leaky wooden dams are woody structures installed in headwater streams that aim to reduce downstream flood risk through increasing in-channel roughness and decreasing river longitudinal connectivity in order to desynchronise flood peaks within catchments. Hydrological modelling of these structures omits sediment transport processes since the impact of these processes on flow routing is considered negligible in comparison to in-stream hydraulics. Such processes are also excluded on the grounds of computational expense. Here we present a study that advances our ability to model leaky wooden dams through a roughness-based representation in the landscape evolution model CAESAR-Lisflood, introducing a flexible and representative approach to simulating the impact of leaky wooden dams on reach and broader catchment-scale processes. The hydrological and geomorphological sensitivity of the model is tested against grid resolution as well as a variability in key parameters such as leaky dam gap size and roughness. The influence of these parameters are also tested in isolation from grid resolution, whilst evaluating the impact of simulating sediment transport on computational expense, model domain outputs and internal geomorphological evolution. The findings show that simulating sediment transport increased the volume of water stored in the test reach (channel length 160 m) by up to an order of magnitude whilst reducing discharge by up to 31% during a storm event (6 h, 1 in 10-year event). We demonstrate how this is due to the leaky dam acting to induce geomorphic change and thus increasing channel roughness. When considering larger grid resolutions, however, our results show that care must be due to overestimations of localised scour and deposition in the model and that behavioural approaches should be adopted when using CAESAR-Lisflood in the absence of robust empirical validation data.

## 1. Introduction

Natural flood management (NFM) seeks to emulate natural processes to reduce flood risk through the attenuation of water, 'slowing the flow' by desynchronising tributaries, reducing surface runoff and/or improving channel-floodplain connectivity (SEPA, 2015; Burgess-Gamble et al., 2017; Lane, 2017). NFM is becoming increasingly popular with flood risk managers due to its multiple benefits and perceived low risk, however due to altering the hydrological regime, there is potential for structures to become displaced and washed out (e.g., Nisbet et al., 2015). NFM is also an effective method to engage local communities and land users in potentially reducing flood risk (Burgess-Gamble et al., 2017; Dadson et al., 2017; Newson et al., 2021). Reintroduction of wood to the river

channel is a popular form of NFM, employed for multiple co-benefits such as habitat creation and ecological enhancements (e.g., Wohl, 2017; Ockelford et al., 2024) as well as flood peak reduction (e.g., van Leeuwen et al., 2024; Villamizar et al., 2024). As a result, NFM now accounts for approximately 20% of UK river restoration projects (Cashman et al., 2018; Grabowski et al., 2019).

One method of introducing wood is through building leaky wooden dams (LDs). LDs are a form of in-channel blockage that can be installed either within a river channel (Metcalfe et al., 2017; Deane et al., 2021) or, as a runoff attenuation feature (RAF), intersecting surface runoff pathways (Nicholson et al., 2012; Nicholson et al., 2019) in an effort to reduce flow velocity and reduce flood risk, increase biodiversity, and improve river heterogeneity (Burgess-Gamble et al., 2017). LDs aim to emulate natural woody debris found in river channels by partially or completely blocking the channel to accelerate the recruitment of natural wood as part of the natural wood cycle (Gregory et al., 1985; Addy and Wilkinson, 2016). LDs have multiple benefits including (but not limited to): improving water quality, increasing habitat diversity, flood wave attenuation, and increasing floodplain connectivity (Wenzel et al., 2014; Burgess-Gamble et al., 2017; Grabowski et al., 2019).

Despite their rapid deployment in riverine management over recent years, a key knowledge gap is how LD efficacy evolves temporally, both in response to geomorphic evolution up- and downstream of the LD, but also in response to flood sequences (Addy and Wilkinson, 2019; Grabowski et al., 2019). The influence of large wood on river systems is well understood: wood increases fluvial complexity whilst being resistant to transportation and providing storage space for water through increasing floodplain connectivity and creating out of bank storage (Gurnell et al., 2018; Wohl et al., 2019). Specifically, large wood can form pools (e.g., Abbe & Montgomery, 1996; Al-Zawaidah et al., 2021; Ravazzolo et al., 2022), increase sediment storage (e.g., Comiti et al., 2008; Wohl & Beckman, 2014), protect against or induce bank erosion (e.g., Abbe et al., 2018; Galia et al., 2024) and influence floodplain morphology (e.g., Sear et al., 2010; Wohl, 2013). Large wood is generally mobile (Wohl et al., 2023), unlike LDs that are often engineered and anchored in-situ and therefore can be functionally different with wide-ranging designs (Lashford et al., 2022; Lo et al., 2022; Quinn et al., 2022).

Challenges in disentangling the relative impact of LDs from the influence of land use, antecedent conditions and other flood risk management interventions presently result in an unclear understanding of their influence over time. Similar to natural wood, LDs influence the hydraulic regime through increasing roughness and thus have the potential to influence channel geomorphology. The few empirical field studies that have focussed on LDs have highlighted that LDs can reduce peak flows for the 1-year annual exceedance probability (AEP) by 10% on average, however the response can be highly variable (van Leeuwen et al., 2024); Norbury et al. (2021) reported an average reduction in peak discharge of 27.3%. The backwater rise induced by LDs is also variable and can be increased or decreased with the presence of porosity-reducing material (Muhawenimana et al., 2023). Furthermore, the ability of a LD to store water, or sediment, can be dependent on the distance between the riverbed and the bottom of the LD, with gaps >0.3 m unable to store sediment in the Yorkshire Dales, UK (Lo et al., 2022), while increased wood volume also amplifies scour (Schalko et al., 2019). Laboratory experiments have shown that representing LDs as non-porous structures increases drag and flow area (Muhawenimana et al., 2021), and therefore it is important to account for porosity of the structures in numerical simulations. Yet often porosity is

not considered in numerical simulations due to representing these complex structures in reduced-complexity models.

Recent works have focused on integrating LDs into 1D and 2D models at different spatial scales (Hill et al., 2023), most commonly representing the interventions as localised roughness adjustments (Pinto et al., 2019; Geertseema et al., 2020), geometry adjustments (Pearson, 2020; Walsh et al., 2020), or a combination of the two (Dixon et al., 2016; Senior et al., 2022). LDs have also been represented in hydraulic models, through stage-discharge relationships realising LDs (and other RAFs) as weirs or culverts (Thomas and Nisbet, 2012; Metcalfe et al., 2017;

Keys et al., 2018; Hankin et al., 2019; Pinto et al., 2019; Hankin et al., 2020; Leakey et al., 2020; Pearson, 2020; Follett and Hankin, 2022). A comprehensive review of the large wood numerical modelling literature focused on artificially placed wood can be found in Addy and Wilkinson (2019).

The vast majority of numerical models used for LD evaluation have not considered the impacts of sediment

transport on function and efficacy. This is in-line with operational approaches to modelling flood risk, where sediment transport processes have often been considered as a negligible source of uncertainty (Flack et al., 2019). LDs and large wood clearly can alter local morphology, which in turn can alter hydraulic response through feedback cycles of erosion and deposition (Lo et al., 2021). Despite this those models that solve only for the hydrodynamic component often produce erodibility maps (Hankin et al. 2019; Pearson, 2020), or report the cross-

sectional- or depth-averaged velocity and shear stress components (Bair et al., 2019) on the bed and banks. However, many previous studies have focused on the reach-scale, or small catchments ($< 10$ km$^2$), simulating one or a small number of LDs (Addy and Wilkinson, 2019) in isolation. It is therefore difficult to validate results at larger scales, especially when combined with a greater range of flows, rarer high flow events and increased complexity (Metcalfe et al., 2017). Those that have attempted catchment-scale simulations, such as the network

models of Hankin et al. (2020) and Follett and Hankin (2022), have not considered sediment transport in any of the scenarios explored. A few studies do exist that simulate sediment transport and riverine geomorphic evolution in response to LDs. Walsh et al. (2020) used the landscape evolution model (LEM) CAESAR-Lisflood (Coulthard et al., 2013) to assess the impact on channel response and suspended sediment discharge of large wood in a small headwater catchment. Large wood was represented using the bedrock layer in CAESAR-Lisflood (i.e., an

unerobible fixed bed) but such an approach does not represent LD function and does not permit the throughflow of water nor represent a lower gap to allow unimpeded passage of baseflows. Pearson (2020) also used CAESAR-Lisflood to implement runoff attenuation and used an approach that features as edits LDs within the terrain. This method allowed features to be eroded, but again lacked porosity and a lower flow gap. As such, no work currently exists that incorporates both the inherent 'leakiness' of LDs and the ability to simulate a lower gap coupled with

sediment transport to evaluate geomorphic evolution within a numerical model. Here we address this methodological gap to advance substantive understanding.

The aim of this paper is to explore the relative behavioural impact of a simple LD on sediment transport processes and subsequent influences on discharge and water storage through a small reach. To do this, we first introduce

new functionality for CAESAR-Lisflood that can represent LDs through the restriction of flow. Second, we evaluate the sensitivity of the model to DEM resolution, and third, assess the impact of LD roughness and gap

size on geomorphology and water storage. Finally, we present the implications of numerically simulating LDs coupled with sediment transport processes to inform future modelling studies.

## 2.  CAESAR-Lisflood

### 2.1.  Model description

Geomorphic processes are complex, and consequently high-fidelity numerical models designed to simulate them are also complex and computationally demanding, meaning long-term simulations (10–100s of years), or multiple simulations of different scenarios, can take substantial computational resources. Time is a barrier to decision makers who may wish to use information from simulations in order to plan flood management interventions and/or

river restoration schemes. LEMs reduce complexity by simplifying processes, increasing computational efficiency and enabling useful and timely information to be extracted. Originally designed to investigate broad scale controls and behavioural changes to landscapes as they develop over long timescales ($10^2$–$10^6$ years), LEMs have been key to a range of advances in the understanding of long-term geomorphic processes. Developments in computational power has increased the complexity of some LEMs whilst retaining their efficiency, leading to the

development of 'second generation' LEMs such as CAESAR-Lisflood, herein referred to as CL. This has extended the capabilities of the original LEM for wider applications, including for example, landslide risk (Xie et al., 2022), hazards to electricity transmission towers (Feeney et al., 2022), mining (Hancock et al., 2017) and flood risk management (Croke et al., 2016; Ramirez et al., 2020).

CAESAR-Lisflood is a second-generation LEM that merged the original CAESAR LEM with the 2D hydraulic code, Lisflood-FP (Bates et al., 2010), replacing the original simplistic steady-state hydraulic code (Coulthard et al., 2013). The development allows the model to simulate geomorphic processes at event-scale whilst retaining its efficiency. Further developments within CL has enabled application in flood risk management through the ability to apply spatially distributed rainfall within the model domain, allowing for representation of convective

events (Coulthard and Skinner, 2016). As such, CL is a suitable model to further enhance with new tools to simulate NFM approaches, such as LDs. CL can have one or more direct hydraulic source inputs that can be used in both catchment and reach modes, in combination with rainfall. CL requires minimal data (elevation and rainfall or a discharge input) for operation, uses readily available regular gridded DEM data with a range of grid sizes, is open source and highly customisable, and crucially can simulate spatially distributed morphodynamic evolution

utilising up to nine grain size fractions (Meadows, 2014; Hancock et al., 2015; Pearson, 2020; Walsh et al., 2020). Fluvial erosion and deposition are governed by three selectable sediment transport laws: Wilcock and Crowe (2003), Einstein (1950) or Meyer-Peter and Müller (1948). As CL ingests a regular raster grid, attributes can be assigned to each unique cell including roughness (Manning's $n$), TOPMODEL $m$ value and more (Li et al., 2023). Here, version 1.9j, first released in August 2019, is used as the baseline for development (available here

https://sourceforge.net/projects/caesar-lisflood/files/).

### 2.2.  Leaky dam module

The approach developed herein represents the leakiness of a LD, its water depth-dependent impact on the water column, and changing efficacy due to implicit geomorphic changes. There are multiple different designs for LDs,

from those that are more natural and better emulate natural large wood, to those that are more engineered, with

slots for water to pass through (Lashford et al., 2022; Quinn et al., 2022). The module presented herein allows the simulation of gaps below LDs, a common design feature, in a way that can alter due to erosion or deposition. In addition, the user can specify an install time within the simulation timeline, thus allowing the model to reach steady state without LDs impacting hydro- or morphodynamics. LDs can then be inserted into evolved landscapes, allowing a range of experimental simulations that more realistically simulate LD installation.


The LD function uses a dynamic value for Manning's $n$ roughness (henceforth $n$) for cells that have been assigned as containing a LD. This method is straightforward to apply within a model domain, as specific cells can be identified to place the LD in combination with other roughness variables such as in-channel or floodplain boundary roughness (Liu et al., 2004; Kitts, 2010; Odoni and Lane, 2010; Dixon et al., 2016; Pinto et al., 2019;

Rasche et al., 2019; Barnsley, 2022; Senior et al., 2022). Roughness values can be determined from field observations and utilised in numerical models (Shields and Gippel, 1995; Curran and Wohl, 2003; Kitts, 2010; Dixon, 2013) but careful consideration of the application and transferability of roughness values between field sites and at different scales must be considered, especially in steep river channels where higher roughness has less impact compared to a physical blockage (Addy and Wilkinson, 2019). There is currently no implementation of a

stage-dependent dynamic roughness value for LDs in the literature. This is an important limitation of previous approaches, since the relationship between flow resistance and LDs is known to be stage dependent (Jeffries et al., 2003; Keys et al., 2018; Addy and Wilkinson, 2019; Muhawenimana et al., 2023). Senior et al. (2022) highlight that care must be taken when interpreting the effectiveness of changing roughness values as it can slow the flow of water rather than discretely store it. The approach adopted herein emulates the behaviours often observed by

LDs, but does not account for the entire hydraulic complexity as obsereved in laboratory studies (e.g., Muhawenimana et al., 2021).

The function applied herein determines the value of $n$ to be used to estimate flow through each cell containing a LD according to the proportion of the water column behind the LD that is in contact with the LD at each timestep. If there is no LD, or the activation criteria are not met, $n$ defaults to the default bed roughness defined by the user,

$n_{global}$. Otherwise, a unique roughness, $n_{local}$ is calculated for each timestep for each LD up to a maximum user defined value ($n_{max}$). Adjustment to $n$ is performed as a function of cell properties (see Figure 1): the initial elevation of the bed ($z_{bed}$); current bed elevation upstream of the LD ($z_{USbed}$); and elevation of upstream water level ($z_{water}$). In addition, there are three user-defined properties: 1) the size of the vertical gap between the river

bed and the base of the LD on installation, $h_{gap}$; 2) the distance between the bed elevation at the start of the simulation and the top of the LD, $h_{top}$; and 3) the maximum value of $n$ if the entire water column upstream of the LD is in contact with it, $n_{local}$.

CL employs a first order upwind scheme (Coulthard et al., 2013). Therefore, for $cell_{xy}$, the model will calculate

elevation values from the cell upwind of the LD from where water originates and the LD is assumed to take effect on the face between cells. The hydraulic model within CL uses the four cardinal neighbours (D4) to transport water and sediment, therefore only connected grid cells can transport material (O'Callaghan and Mark, 1984), resulting in no diagonal connections, as with the D8 flow direction algorithm. As such, for cell properties, the $x$

and $y$ coordinates can vary based on flow direction: $property_x = cell_{xy} \mid cell_{x-1,y}$, depending on which has the greatest water level in the $x$ direction, and $propetry_y = cell_{xy} \mid cell_{x,y-1}$ for the $y$ direction.

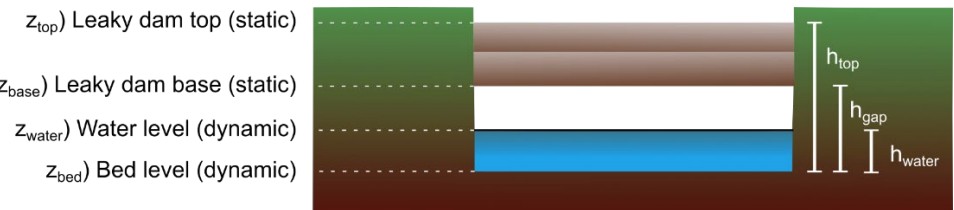

**Figure 1: Schematic showing hypothetical cross-sections for LD-containing cells. Elevations are represented as $z_n$ and heights relative to $z_{bed}$ as $h_n$. The LD becomes effective once the elevation of the water level ($z_{water}$) exceeds the elevation of the bottom of the LD ($z_{base}$). Throughout simulations, the elevations ($z_{base}$ and $z_{top}$) expressing the absolute top and bottom elevations of the LD do not change so changes to the elevations of the water level ($z_{water}$) and the bed level ($z_{base}$) will change its efficacy.**

There are two different methods of assigning a cell as containing a LD. The first method uses codes with each cell assigned a value between 0 and 5. If it is 0 there is no LD, if it is greater than 0, the cell is assigned one of five user-determined LD parameters, including a gap size ($h_{gap}$), height ($h_{top}$), and maximum roughness. Upon initialisation, the model will convert those parameters into $z_{base}$, $z_{top}$, and $n_{max}$. Additionally, the module determines the upstream direction and automatically assigns the LD to the corresponding cell face as shown in Figure 2. This enables the LD to be placed without considering flow direction.

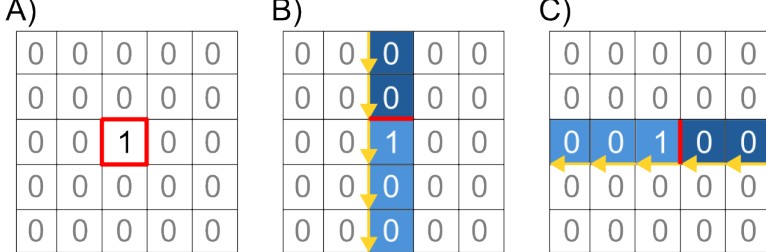

**Figure 2: Leaky dam representation on a regular grid where 1 denotes an LD. Red shows the potential location for the LD along a cell face, yellow the flow direction between cells and blue relative water depth with a darker colour denoting deeper water. The location of the LD changes with flow direction. A) zero flow, therefore the LD could be on any cell face; B) flow from North to South, therefore the LD is on the Northern cell face; C) flow from East to West, therefore the LD is on the Eastern cell face.**

When the model is initialised, for each timestep the LD module will run through an iterative process, represented by Equation (1), that will determine the proportion of the LD in contact with the water column and calculate the scale of $n$ using a blockage ratio ($BR$). BR captures increasing LD roughness with increasing stage to reflect increasing complexity as a greater vertical area of the channel is obstructed. Further empirical data is required to assess this assumption. If the LD is overtopped, the blockage ratio reduces because the cross-sectional area of flow increases while the cross-sectional area of the LD remains constant. Therefore the relative cross-sectional roughness reduces and thus $n$ is reduced. Equally if the LD is not at maximum capacity, that is $h_{water} = h_{top}$, $n$ will is scaled to less than $n_{max}$ accordingly. The global Manning's value ($n_{global}$) is then combined with the scaled $n_{max}$ to create $n_{local}$ as shown in Equation (2) which is used in subsequent processing steps by CL.

$$1) \qquad BR = \max\left(\frac{\min(z_{water}, z_{top}) - \max(z_{base}, z_{bed})}{h_{water}}, 0\right)$$

$$2) \qquad n_{local} = n_{global}(1 - BR) + n_{\max}BR$$

The proposed depth-weighted roughness representation method enables the user to emulate real-world implementations of LDs. Roughness (or porosity) quantification for structures such as LDs is often impractical at a large scale due to the required resolution of remotely sensed data, as well as the characteristics of the LD itself—geometry, litter cover, sorting, and wood size for example (Dixon, 2016; Livers et al., 2020). As such, the $n$ method represents a range of values that can be used to assess the extent of flow restriction caused by an LD and its relative impacts.

## 3. Methods

Prior to evaluating the impact of LDs on the hydrogeomorphology, sensitivity tests were conducted to understand the relationship between DEM grid resolution and the impact of simulating sediment transport compared to only the hydraulic component. Sensitivity analysis was performed using a single rainfall input—a six-hour, 10% AEP rainfall event—derived from the flood estimation handbook (Stewart et al., 2013) on a synthetic DEM (herein referred to as $DEM_T$).

### 3.1. Synthetic reach-scale terrain

The model domain was 160 m long and 100 m wide and represents a second-order stream. The DEM had the same average slope as a prototype site (0.01 m m$^{-1}$; Wolstenholme, 2023) where LDs were installed in 2019 and was created by linear interpolation between the high and low survey points in the reach captured with a Topcon OS-103 Total Station (TS). $DEM_T$ was resampled preserving minimum elevations to $DEM_{Tj}$ where j represents the cell resolution (of either 1, 2 or 4 m as part of the grid sensitivity tests), ensuring channel depth and slope angle were preserved. To assess the influence of DEM resolution on model behaviour, a 1 m-deep, 4 m-wide channel was burnt into all DEMs.

### 3.2. Model set-up

A nested approach was used to drive model experiments at the reach scale. First, to derive discharge and sediment input for the reach of interest, discharge from the wider upper catchment, $DEM_C$ (2 km$^2$; obtained from OS Terrain 5 data at 5 m resolution; Ordnance Survey, 2020), was modelled using an extract of radar rainfall observations derived from the UK NIMROD radar network record for 2006–2020 (Met Office, 2003). This was applied at a 60-minute timestep to $DEM_C$ as a global rainfall input to spin up the model and remove initial sediment extremes exported from the system due to an initial condition of homogenous grain size distributions across the DEM. This ensured that the sediment was distributed throughout the catchment in equilibrium with the topography. This was then repeated to derive a hydraulic and sediment discharge input for the reach scale model.

The input grain size distribution was calculated using field data from Wolstenholme et al. (2024). The site from which the grain size distribution was collected was approximately 2 km downstream of the LD, because LDs had been installed prior to surveying as the channel grain size distribution in the reach of interest would not be representative of a pre-LD scenario. The b (intermediate) axis of >400 randomly selected clasts were measured from four locations in the reach, and the distributions binned into the nine default classes, as used by CL (see Figure 3). The grain size distribution was found to have a $D_{50}$ of 12.8 mm, which was applied globally across the modelled reach domain. Within the model no sediment was transported in suspension and the Wilcock and Crowe (2003) sediment transport law was selected within CL since it was developed using a mixture of both sand and gravels, which is appropriate for the grain sizes used.

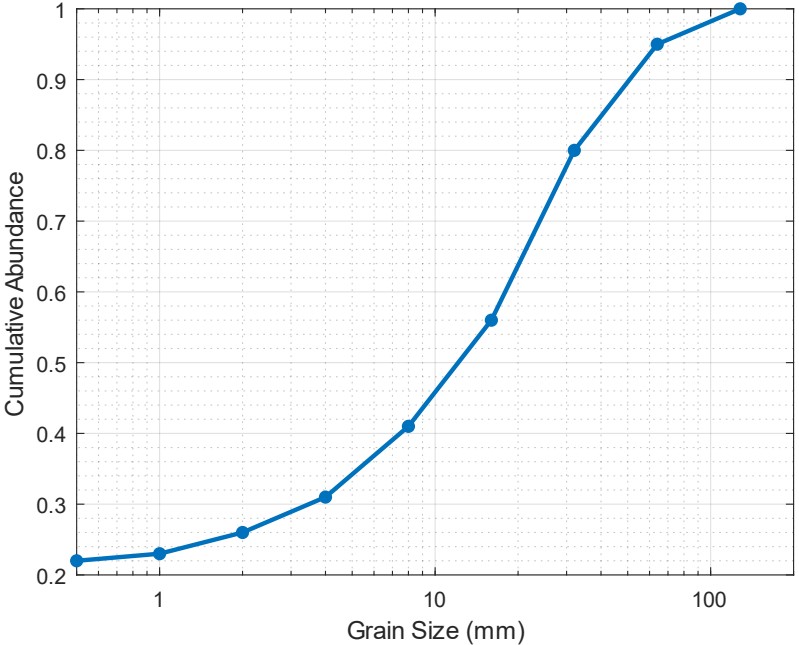

**Figure 3: Grain size distribution used in experiments from Wolstenholme et al. (2024).**

### 3.3. Experimental design

Each $DEM_{Tj}$ had a single LD installed 100 m downstream of the model input to assess geomorphic effects surrounding the LD and reduce potential impact from the model boundaries. A single rainfall storm was used (0.1 AEP with a six-hour rainfall duration) nested within a 120-hour period of baseflow with a 60 second timestep. An initial period of 33.3 hours was used to fill the river reach and establish a hydraulic equilibrium, followed by the input to the reach model of a catchment-derived storm for a further 50 hours, then baseflow for the remainder of the run. The storm input was appended onto the spin-up period to ensure that all experiments that involved sediment transport had identical initial conditions. The LD was "installed" following the spin period, 33 hours prior to the onset of the storm used for analysis. This allowed the river channel to adjust to baseflow without being impacted by the LD. Two LD gap variants were tested in each scenario (0 m and 0.2 m), and the maximum roughness ($n_{max}$) of the dams was set to 0.16 (chosen as a conservative estimate of LD roughness after Curran and Wohl, 2003; Dixon et al., 2016; Addy and Wilkinson, 2019 ).

LD height and width was constant throughout the experiments (0.5 m above initial channel bed, 4 m wide). Output data from the simulations, which provided information on outlet discharge, sediment yield for each grain size and total sediment yield, was recorded at a one-minute timestep. All tests were repeated with CLs 'flow only' option. When on, the erosion and deposition modules of the model are bypassed, and it functions as a 2D hydraulic model with a rigid boundary, as such hydraulic simulations were started 30 days prior to the onset of the storm. To assess the impact of the LDs on the system, the difference in peak discharge and storage capacity over time were calculated, comparing the cumulative discharge for a given storm to a corresponding 'no LD' baseline scenario. Finally, the influence of changing LD roughness and gap size was assessed.

## 4. Results

### 4.1. Computational expense

Figure 4 shows the number of iterations required to complete the simulations for the performed experiments. Iterations are a useful proxy for model efficiency, independent of the computer used to perform the simulations. They show that for decreasing grid resolution, fewer iterations are required across all experiments. Simulation of the LD with only the hydraulic model enabled results in little increase in computational expense (averaged standard deviation = 20,259) whereas enabling sediment transport drastically increased model iterations often by over 200% across all DEM resolutions.

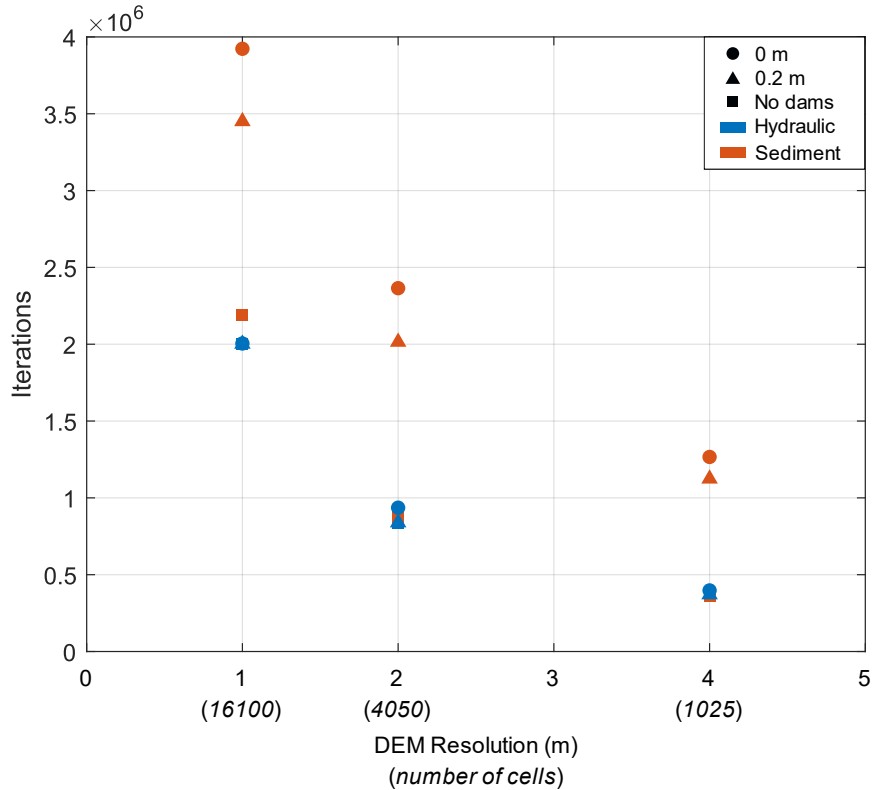

**Figure 4: Computational expense represented as the number of model iterations for simulations with, and without, LDs.**

### 4.2. Model domain outputs

The influence of the LD on the model domain outputs was assessed through comparing the baseline (no LD scenario) to each of the LD variation experiments after the spin period, measured from the outlet of the model. Hydraulic-only simulations were more stable than the sediment transport enabled counterparts as shown in Figure 5. When $h_{gap} = 0$, hydraulic-only experiments show discharge attenuation of up to 1% when the LD is installed and further attenuation of up to 2.4% on the rising limb of the storm, before increasing $\Delta Q$ (the change in $Q$ between simulations) to 3.1% during the peak. When $h_{gap} = 0.2$, instantaneous discharge attenuation is not seen when the LD is installed, but the LD did increase discharge on the rising limb and the peak by up to 3%, before attenuating $Q$ on the falling limb for both higher DEM resolutions, but reduced flows when a 1 m grid cell size is used (Figure 5).

In contrast, when sediment transport was enabled $\Delta Q$ was more variable after the storm. There was instant attenuation of up to 50% when $h_{gap} = 0$, for $DEM_{T2}$ and $DEM_{T4}$, and also up to 50% attenuation during the storm (see Figure 5C). In contrast, when $h_{gap} = 0.2$, there was no reduction in $Q$ upon LD installation as seen in the equivalent hydraulic-only scenario, and reduction in $Q$ of up to 4.7% for the 4 m DEM during the rising limb. For all sediment transport enabled experiments, the falling limb of the storm and remainder of the simulation time shows up to ±25% deviation from the baseline scenario due to $Q$ becoming out of phase with the baseline. $DEM_{T1}$ had an overall reduction in $Q$ (maximum = –10%) following the peak of the storm compared to coarser grid resolutions $DEM_{T2}$ and $DEM_{T4}$ that increased $Q$ (maximum +12% and +25%, respectively). Where $h_{gap} = 0.2$, although the influence of the LD had little impact on the peak, there was much larger disruption to the discharge

on the falling limb. These disruptions represent a deviation in $Q$ of up to 0.014 m$^3$s$^{-1}$ as a result of sediment transport and the presence of the LD.

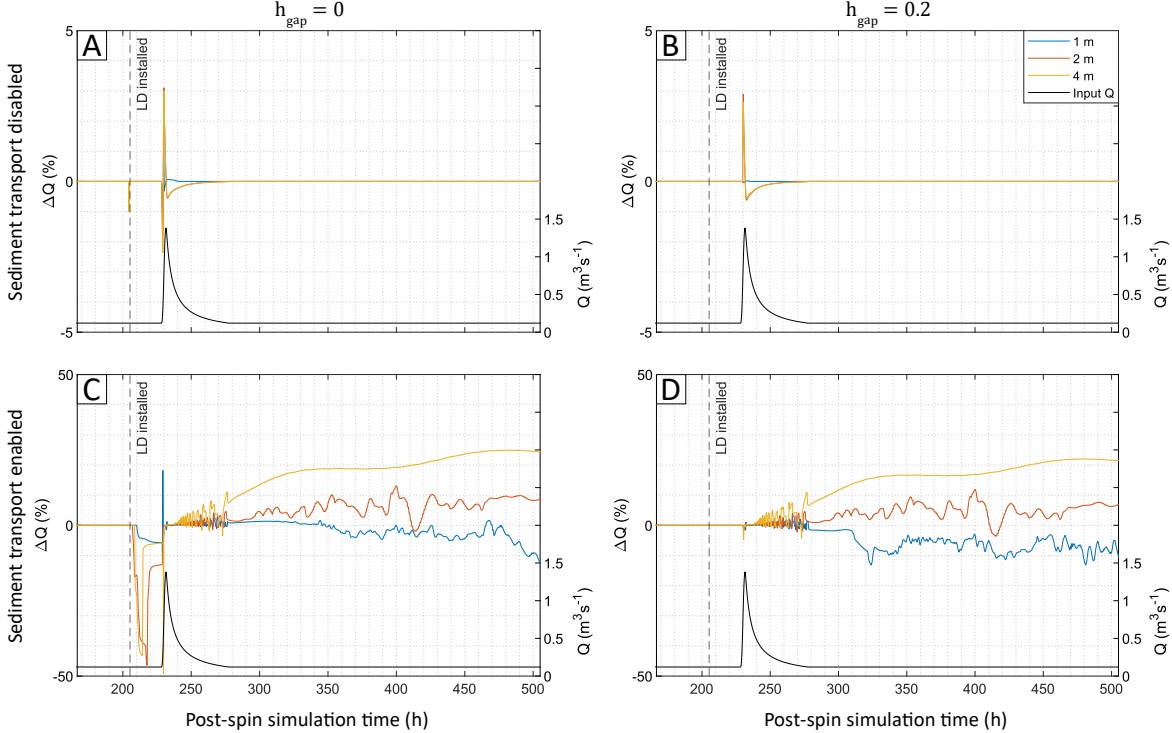

**Figure 5: Impact of DEM grid resolution on predicted $Q$ for sediment transport disabled/enabled experiments with LD gap size set to 0 m (A, C) or 0.2 m (B, D). Black line denotes input discharge (right axis, uniform across all experiments).**

The impact on $Q_s$ (sediment discharge) was determined using the same approach for $Q$ detailed previously, but by comparing the cumulative $Q_s$ rather than instantaneous discharge. When the LD was installed, there was no impact on $Q_s$ efflux of the model domain for any experiment. Across all the grid resolutions used, $\Delta Q_s$ was found to increase immediately following the peak of the storm and sediment was lost from the system. The 4 m grid resolution had a substantially larger sediment efflux (see Figure 6) when $h_{gap} = 0$ and to a lesser degree where $h_{gap} = 0.2$. On the falling limb and remainder of the simulation the 1 m and 2 m DEM resolutions had negative $\Delta\Sigma Q_s$ indicating that sediment efflux was lower than that of the baseline scenario and sediment was stored in the reach. Although the 4 m resolution showed a similar behaviour, sediment efflux was continuously greater than the baseline.

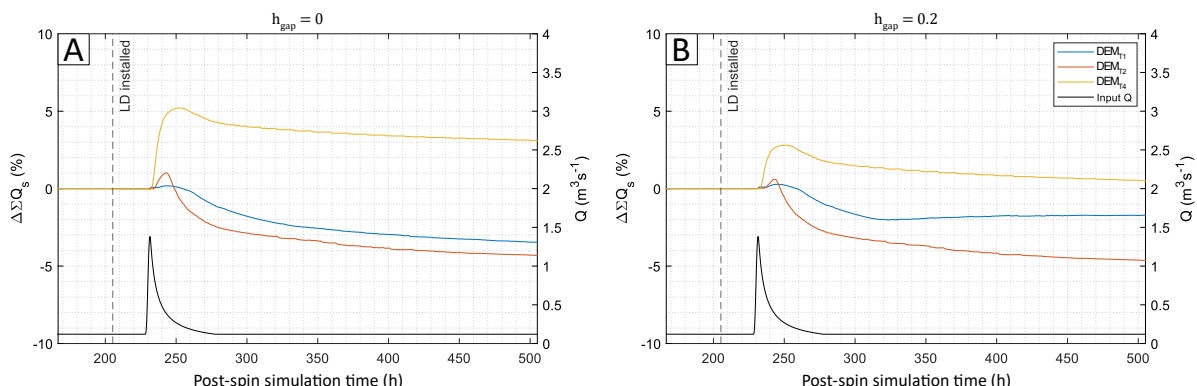

 **Figure 6: Cumulative sediment discharge ($\Sigma \Delta Q_s$) for 0 m LD gap (A) and 0.2 m gap (B) experiments. Input discharge shown for reference.**

### 4.3. Geomorphological evolution

Geomorphological evolution was assessed within the model domain by comparing the average channel width elevation change to the baseline scenario for all experiments shown in Figure 7. Installing a single LD substantially

influences bed elevation change throughout the system. All simulations regardless of grid resolution follow the same pattern of elevation change. There was increased deposition 60–100 m downstream (average +0.12 m, maximum +0.25 m). The cell immediately upstream of the LD was typically erosive for $DEM_{T4}$ when $h_{gap} = 0$ and 0.2 (0.07 and 0.11 m respectively). When the channel width was greater than one cell (i.e., $DEM_{T1}$ & $DEM_{T2}$) there was also deposition predicted in the upstream cell of up to 0.12 m. Typically when $h_{gap} = 0.2$ there was

less bed erosion predicted.

Immediately downstream of the LD had the most substantial bed elevation change, with all scenarios being highly erosive from -0.08 m ($DEM_{T1}$; $h_{gap} = 0.2$) to -0.99 m ($DEM_{T4}$; $h_{gap} = 0$). When $h_{gap} = 0$ there was more erosion in the downstream cell than when $h_{gap} = 0.2$. Downstream of the LD a depositional zone was predicted,

with an elevation change similar in magnitude to the eroded cell upstream, ranging from 0.2–0.85 m ($DEM_{T1}$; $hgap = 0.2$ and $DEM_{T4}$; $h_{gap} = 0$ respectively), and thus similarly there was more deposition where $h_{gap} = 0$. Overall, the order of magnitude and directionality of the elevation change is similar across all scenarios. Finally, for all scenarios following the second cell most downstream of the LD there was fluctuating propagation of bed elevation change predicted of the order of ±0.10 m until the edge of the model domain.

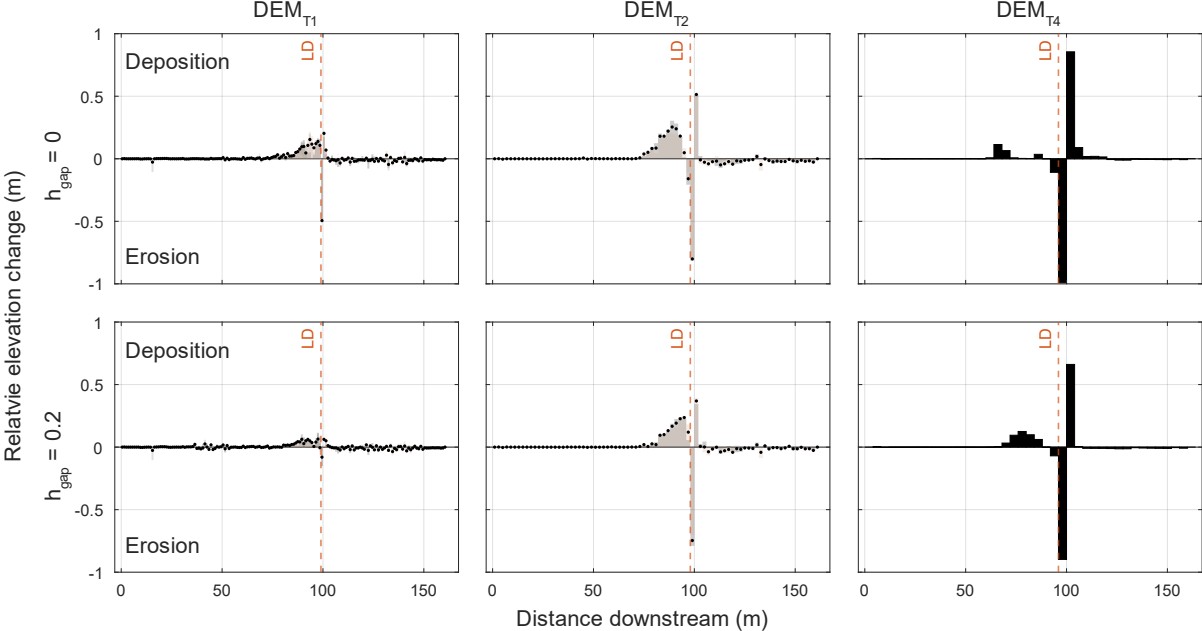

**Figure 7: Relative average channel width bed elevation change for sediment transport enabled experiments. Bars show individual cell elevation change compared to the baseline scenario with the mean represented as a point for DEM_{T1} and DEM_{T2}.**

### 4.4. Leaky dam parameter adjustment

To explore the impact of varying $n$ and the LD gap to build an envelope of potential responses, DEM_{T2} was chosen to use the highest resolution DEM that is practical to run across a suite of scenarios, whilst retaining a reasonable representation of the initial topography. Grain size, rainfall input and LD location were kept constant. Using the same hydraulic input as above, a no-LD baseline experiment was performed in addition to a matrix of 25 tests varying two LD parameters. First, the maximum LD roughness ($n_{max}$) was varied between 0.12–0.20 s m$^{-1/3}$ at intervals of 0.02 s m$^{-1/3}$. Values herein are based on empirical studies and are representative of naturally occurring log jams (Curran and Wohl, 2003; Dixon et al., 2016; Addy and Wilkinson, 2019). Second, the LD gap size was systematically varied from 0–0.4 m at 0.1 m intervals. Hydraulic and sediment transport enabled simulations were both performed, resulting in a total of 52 experiments.

### 4.4.1. Water storage

Sediment transport enabled simulations showed at least an order of magnitude greater water storage than the hydraulic equivalent. Where there was no LD gap, water was instantly stored upon LD installation until the onset of the storm where rougher LDs were found to store the greatest volume of water. Larger gap sizes did not store water upon LD installation, rather when stage reached the base of the LD. Where the gap size was 0.3 m or greater, there was no difference in water storage and therefore the LD did not engage with the river. For all remaining scenarios, water storage was greatest during the peak of the storm, with increasing gap size resulting in diminished storage during the peak for both hydraulic and sediment simulations. There was no subsequent water storage for hydraulic simulations except where there was a 0 m gap. In contrast when simulating sediment transport, the system stored 0.95–2.2 m$^3$ of water when compared to the baseline experiment, as shown in Figure 8, with diminishing effectiveness when gap size was increased.

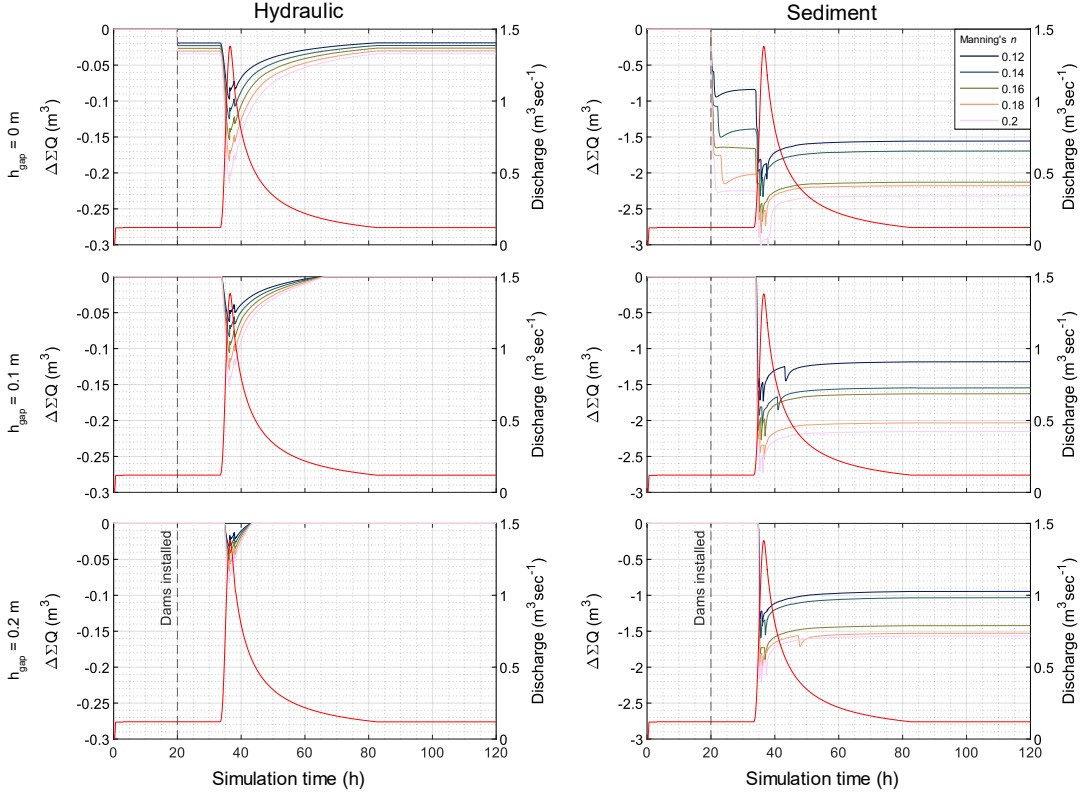

**Figure 8: Water storage ($\Delta\Sigma Q$) for hydraulic and sediment transport enabled simulations, separated by LD gap size in metres (0.3 m and 0.4 m gap simulations were omitted due to their having no impact). Manning's $n$ (s m$^{-1/3}$) denoted in legend with input discharge represented in red. Note different y-axis scale.**

### 4.4.2. Sediment transport

Sediment transport was found not to be influenced by the installation of the LD at 20 hours, regardless of gap size, however only when the gap was 0.2 m or less did the LD influence $Q_s$ during the storm (see Figure 9). Prior to the peak of the storm, $Q_s$ was reduced by <0.03 m$^3$ when the LD engaged with the flow compared to the baseline scenario (see Figure 9A'–C'), but there was little variability prior to the peak (average standard deviation: 0.008). Following the peak of the storm, the influence of roughness variability was more pronounced with an increase in $Q_S$ ($\Sigma\Delta Q_s$ 0.25–0.8 m$^3$), however there was no clear trend between the volume of sediment exiting the model domain (Figure 9) and the chosen roughness value. However, increasing LD gap size did result in less sediment being lost out of the domain, likely due to there being a lower energy gradient induced by the LD. Following the storm, sediment discharge was lower than the baseline scenario (except where $n$ was 0.14 or 0.18 s m$^{-1/3}$ and there is no LD gap) with a maximum difference of 0.5 m$^3$ (where $n$ is 0.16 and the LD gap is 0.2 m).

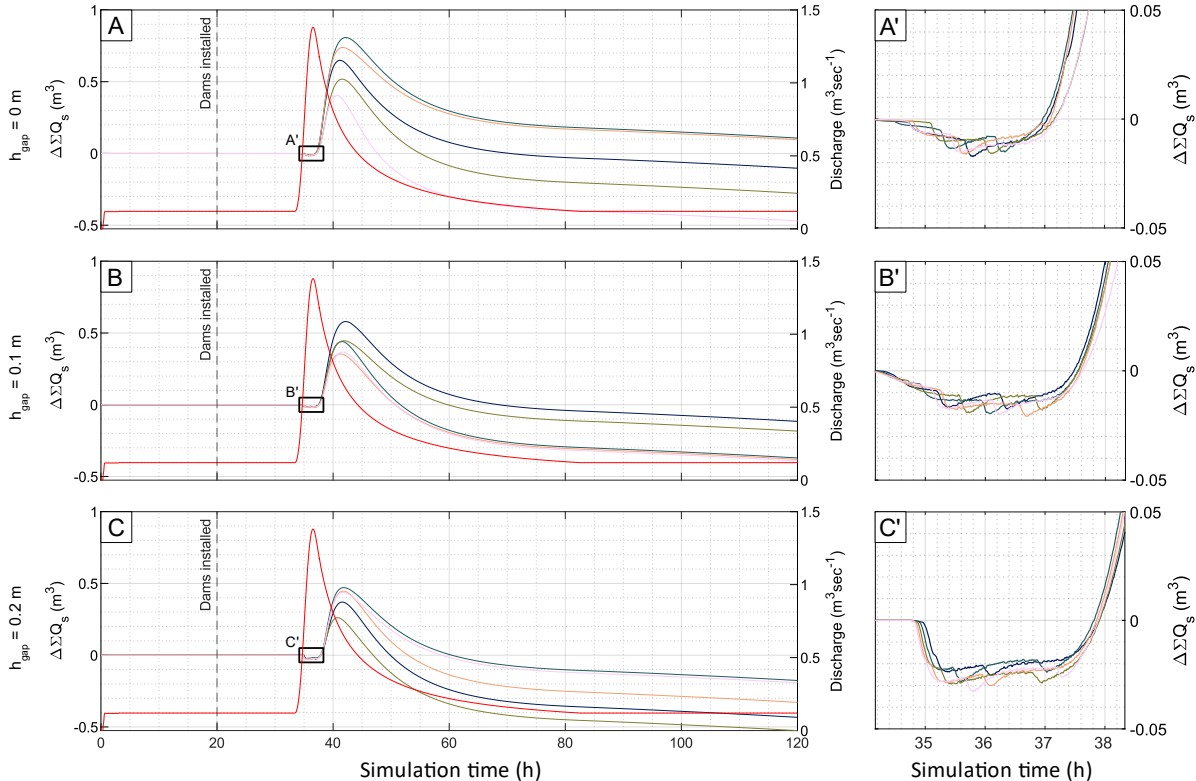

**Figure 9: Cumulative sediment yield difference compared to the baseline scenario for 0, 0.1 and 0.2 m gap sizes (A–C respectively) separated by Manning's n (s m$^{-1/3}$). Input discharge shown by red curve. 0.3 m and 0.4 m gap simulations were omitted due to their having no impact.**

### 4.4.3. Elevation change

In section 4.3 it was shown that the LD mainly impacted the elevation of cells immediately upstream and downstream of the LD. As such, only these three sections of the channel were considered for localised bed elevation change analysis, as shown in Figure 10. When the LD gap was 0.3 m or greater, there was no influence on bed morphology. Immediately upstream of the LD (Figure 10A) bed erosion magnitudes of 0.05–0.14 m were simulated, with a 0 m gap and high roughness (0.18–0.2 s m$^{-1/3}$) LD conditions, with lower volumes of erosion simulated with larger gap sizes and lower roughness values. There was a non-linear relationship between these parameters, with erosion magnitudes decreasing rapidly for higher roughness values as gap size increases, yet not as rapidly with lower roughness values applied. For example, where the LD gap is 0.1 m, erosion magnitudes are more closely clustered (0.105–0.092 m) than for other gap sizes. Zones immediately downstream of the LD (Figure 10B) experienced the most erosion, with up to 0.85 m of scour. The patterns and relative levels of bed erosion closely matches that of the zones immediately upstream of the LD, however at a greater orders of magnitude. Finally for the second cell downstream (i.e., 4 m from the LD), there is only deposition predicted (0.35–0.57 m). Higher roughness values experienced greater levels of deposition, which also decreased with increasing gap size.

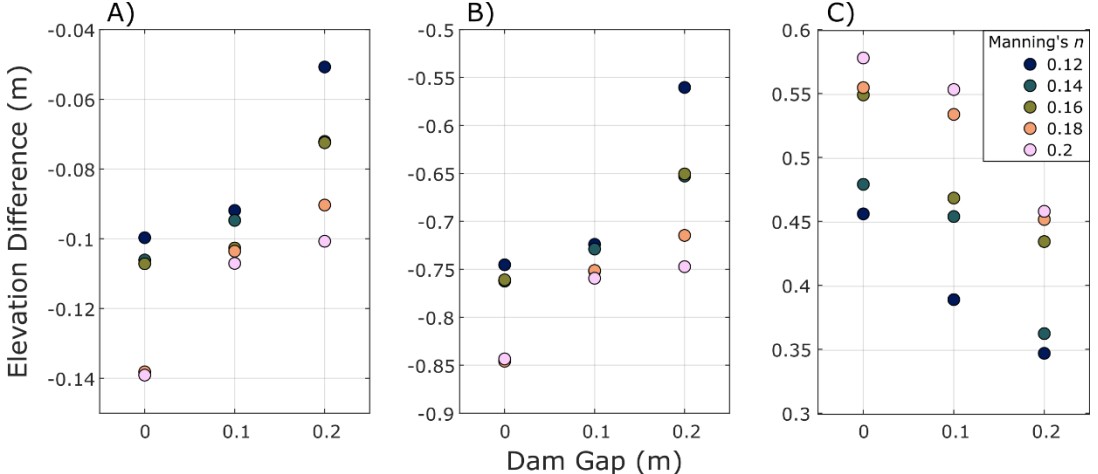

**Figure 10: Elevation change for the cell immediately upstream of the LD (A), immediately downstream (B) and 4 m downstream (C) separated by Manning's n (s m$^{-1/3}$). 0.3 m and 0.4 m gap simulations were omitted due to their having no impact.**

## 5. Discussion

### 5.1. The CAESAR-Lisflood model

The results presented above demonstrates how CL can effectively represent LDs by dynamically adjusting localised Manning's $n$ roughness values whilst accounting for different LD heights and gap sizes as a function of the proportion of the water column upstream of the LD. Placement of LDs is straightforward and can be achieved using a geographic information system, or through creation of a regular grid file, and simulations can be achieved using minimal data—a DEM, hydraulic input, and sediment GSD—to produce an overview of how a reach responds to LD installation. CL could therefore be used to explore the relative influence of LDs and their parameters throughout a given reach to indicate a behavioural response, such as increased flow attenuation or enhanced geomorphic diversity following calibration. Through developing the ability to install a LD after a river channel has evolved to baseflow conditions, the simulation, and its output, are less affected by the bias of having these structures installed from the start of the simulation. As such, LDs are generated after baseflow has been established, which is more representative of the real LD installation process where wood is often felled and then anchored *in situ* while the river is flowing (Grabowski et al., 2019; Lo et al., 2021).

The effect of LDs—as well as natural log jams—is known to be stage dependant (Jeffries et al., 2003; Keys et al., 2018; Addy and Wilkinson, 2019; Muhawenimana et al., 2023), therefore it is important to ensure that the model can account for Manning's $n$ being variable. Development of the LD module introduced above in CL builds upon previous work outlined by others (Dixon et al., 2016; Addy and Wilkinson, 2019; Hankin et al., 2020; Pearson, 2020; Walsh et al., 2020; Hill et al., 2023), filling a research gap through provision of a tool capable of generating dynamic roughness values that ensure that temporal variability in the stage-dependent LD-water relationship is captured. In addition, as CL utilises a regular raster grid, the required resolution can be adapted based on user requirements as well as computational resource availability. CL is a reduced complexity LEM that is not designed to simulate complex high-resolution channel flows, rather larger reach- and catchment-scale change over longer durations. As CL has the capability to simulate high-resolution environments (both spatially and temporally) at

increased computational expense, it is important to recognise the impact of cell size throughout simulations. However it is also important to note in the simulations presented above that the order of magnitude of predicted change was similar, regardless of model cell size, for water storage, geomorphology and discharge.

High-resolution cell sizes are the most computationally expensive. Increasing spatial resolution from two metres to one metre results in a four-fold increase in the number of cells that occupy the same extent, with almost double the number of model iterations. The increase in iterations is also dependent on the area that water interacts with, and the impact the water has on those cells. When simulating the impact of LDs and sediment transport, there was a two-fold increase in number of iterations compared to a no-LD scenario at all resolutions tested. In addition,

higher resolution cell sizes (such as two metres) without LDs performed a similar number of iterations to the five metre resolution experiments with LDs when simulating sediment transport. As such, scale has substantial consequences for future work when simulating large catchments, despite having minimal influence on discharge. Increasing cell size results in a decrease in the accuracy of the true topography as the landscape is smoothed (Schoorl et al., 2000; Claessens et al., 2005), therefore it is important to utilise as high a resolution of grid as

practical, without compromising the quality of model outputs. Resampling the input DEM can lead to variable channel widths and therefore a vast difference in potential water storage, which can introduce a substantial amount of bias into the results if not considered. Due to this, CL should only be used to understand broad behaviours that might be representative of a reach or catchment to discern information of interest where a finer resolution is impractical, especially at larger scales.


Cell size also has an impact on predicted geomorphic evolution. Higher resolutions can capture smaller scale fluctuations in bed elevation. Nevertheless herein it was found that coarser resolutions were able to predict relative changes that were of similar magnitudes to those predicted for finer resolution grids. All simulations predicted scour immediately downstream of the LD followed by a zone of deposition, with the perturbation fluctuating in

magnitude as the signal weakens distal to the LD. Skinner and Coulthard (2022) showed that in CL as DEM grid cell increases, the representation of the hydrological network can become degraded, and although the model recorded similar total sediment yields following a 30-year continuous time series over a 0.5 km$^2$ catchment, this was from fewer geomorphologically active events. In this study a single-thread linear channel is used to evaluate the behaviour of the LD without introducing more complex morphological change or alterations to flow

characteristics, similar to a laboratory environment. As such, the findings of Skinner and Coulthard (2022) regarding connectivity do not apply here, however further testing is required to evaluate the impact of a longer time series on geomorphic evolution and LD efficacy.

LEMs are notorious for being difficult to validate due to the lack of availability and paucity of calibration data

(Wong et al., 2021). Combined with many adjustable parameters and initial conditions, there is a high probability for model equifinality (Coulthard and Skinner, 2016; Hancock et al., 2016; Skinner et al., 2018; Skinner et al., 2020). It is, however, possible to treat the outputs from an LEM, such as CL, in more abstract perspectives and use the results to identify the influence of intrinsic variables and the addition of structures to a system. As such, care must be taken when extracting and interpreting data outputs and using appropriate metrics in order to

capitalise on data produced (Skinner et al., 2018). When simulating sediment transport and recording the output

with high temporal resolution, discharge contained sharp increases and decreases due to pulses of sediment being suddenly mobilised within the system when the transport threshold was reached (as seen in Figure 5). Thus, water storage is perhaps a more useful mechanism for accessing the influence of LDs at this scale, especially without producing vast amounts of extra data. A direct comparison can be performed between simulations which indicate the total volume of water being stored in a system compared to a system without LDs installed. The same practice can be applied to sediment stored within the system and aligned with the output hydrograph.

### 5.2. Sensitivity considerations

For the simple reach $DEM_T$ used in the sensitivity suite of experiments it is clear that finer resolution grids are more computationally expensive due to the number of cells to be processed, yet the LD module almost doubles this computational expense. Despite this, elevation change across the river profile follows a similar pattern regardless of grid resolution and also has the same order of magnitude and directionality of change as shown in Figure 7. Grid resolution when simulating LDs impacts both $Q$ and $Q_s$ through increasing the effectiveness of $Q$ reduction for both hydraulic and sediment transport enabled experiments. The activation of the LD also varied by up to 40 minutes depending on grid resolution. When simulating sediment transport, $Q$ is substantially noisier than hydraulic only simulations, most likely due to CL reaching the threshold required to transport sediment. The noise in the data must also be considered when analysing simulation outputs, however, this may also be a function of the high temporal resolution output recording. Averaging sediment transport data to a lower temporal resolution (e.g., hourly) results in smoother outputs, at the cost of temporal detail. An alternative measure, water storage, calculated as the difference between cumulative $Q$ for both the baseline and the LD implementation simulations may provide additional clarity on the broad impact of LD interventions. Additionally, $Q_s$ and sediment storage is drastically different when increasing grid resolution from 2 m to 4 m, resulting in a five-fold increase in sediment discharge.

Care must be taken when using the LD module for CL as the right results, such as elevation change and $Q$ reduction, may be overestimated when using coarser grid resolutions. The scenarios simulated herein align well with the relative influence of LD from both field and laboratory observations including the formation of downstream pools (Lo et al., 2021; Lo et al., 2022; Muhawenimana et al., 2023), and the potential for sediment storage upstream (Comiti et al., 2008). Future work should focus on calibrating and developing this tool as a flexible and rapidly deployable option for LD simulations in CL, that should currently be used heuristically to mitigate the need for calibration. The LD module for CL can therefore be best used to understand the relative impact of LDs in larger, complex catchment to identify their individual impact on FRM.

### 5.3. Implications

Typically, for FRM, interventions are designed to reduce the risk of a specific flood event threshold derived from historic empirical data within a catchment. The effectiveness of an FRM structure in reducing the impacts of a given AEP is established through rigorous hydraulic modelling of different dimensions of the flow and structure, however modelling must be proportionate to the project considered (Environment Agency, 2022). The results herein show that understanding the influence of an FRM intervention on sediment transport is vital as geomorphic forcing resulting from a structure enables the estimation of the efficacy of an intervention for a catchment.

Sediment transport becomes increasingly important when unintended geomorphic adjustment to 'hard engineered' structures reduces the efficacy of structures, potentially increasing flooding downstream (Hesselink et al., 2003; Pinter et al., 2006; Hudson et al., 2008; Benito and Hudson, 2010). Utilising an understanding of both historic and present geomorphic changes to structures enables geomorphologists to inform FRM strategies (Arnaud-Fassetta et al., 2009). Indeed, without understanding the geomorphic consequences, flood mitigation interventions have the potential to do more harm than good.

Channel evolution alongside LD interventions must be considered for both single storm events and long-term simulations due to the observations identified above. An LD fixed *in situ* can have a substantial effect on the hydrological regime as well as the boarder geomorphology of the river channel, which can, in turn, influence outcomes of flood risk modelling. Often, numerical modellers omit geomorphological process—especially at the event scale—for increased computational efficiency as they are considered to not have an impact greater than that of the uncertainties already present within the model (Flack et al., 2019), yet impounding a channel with an LD can cause substantial geomorphological evolution from a single event alone.

It has been shown that modelling sediment transport can have an impact on the total volume of water that a reach is able to store compared to modelling hydraulics alone. It is important to therefore consider how modelling these processes can further inform future works, such as placement of LD interventions throughout a catchment, as well as how best to utilise these resources to effectively identify locations for river restoration projects. Additionally, numerical modellers can utilise the LD module in CL from minutes, event-scale, annual, decadal and greater if desired, customising the outputs to the users' needs. CL has the capability to save an elevation file (amongst many others) at a given timestep, furthering the understanding of how these structures evolve throughout a storm or rainfall sequence. Additionally other types of structures with porosity can be evaluated with the model, if it can be defined through its height, gap (or lack of) and roughness, for example natural wood, bridges and bunds, and presents opportunities for further study.

The results herein suggest that practitioners should carefully consider the LD gap size as well as roughness of the intervention when installing the structure. Results here show that there is little impact on peak discharge for a single LD in a linear system, however the LDs used here are not designed to engage with the floodplain, and it is therefore not utilised. Larger gap sizes activate later in the storm, therefore may be used as a flood delay system to only capture high flows above a certain height, with careful understanding of flow conditions where the LD is installed. Furthermore, adding more roughness elements to the LD increased potential water storage. The CL tool here uses roughness on a relative scale to provide insight into the impact of a rougher and less rough structure. Roughness can be combined with gap size to produce a similar effect, for example for $DEM_{T2}$, a $h_{gap} = 0.1$ and where $n_{max} = 0.14$ had a comparable and similar impact to conditions where $h_{gap} = 0.2$ and $n_{max} = 0.2$ on downstream deposition. The implementation of dynamic roughness also advances simple LD representation in numerical models, particularly when exploring multi-LD reach and catchment-scale scenarios.

Natural flood management practitioners could also utilise the CL NFM tool to provide an understanding of how installing a given number of LDs may impact their reach and/or catchment of interest to develop a "big picture"

overview of their effectiveness for their given application such as, for example, sediment management, flood risk reduction, or habitat development. Numerical modelling should be used in conjunction with field studies to evaluate the potential effectiveness and cost-benefit analysis of the installation of multiple NFM interventions such as LDs. By enabling researchers and practitioners to easily implement LDs into CL, upper and lower boundaries of the potential impact of installations could be calculated and integrated into different climate scenarios if required. CL presents an opportunity to achieve this with minimal data requirements, so long as the user understands that the output should be regarded as a tool to investigate behaviour of the system and not forecasting the definite impact of LDs or other NFM interventions on a river system.

## 6. Conclusions

This study incorporated leaky wooden dams (LDs) into a numerical model capable of simulating both the hydrology and the sediment transport efficiently at the reach scale. The model also has scope to expand to the catchment-scale whilst simulating multiple complex storm events. The approach used herein has shown that it is important to consider sediment transport and morphological evolution when numerically modelling leaky dams, even at event scale. This is because of the impacts this has on altering the total volume of water stored by the LDs, in addition to inducing greater geomorphological complexity. Based on a synthetic DEM, LD gap size was shown to be much more important than dam roughness when numerically modelling with the CL method and could be utilised in the future by NFM practitioners looking to install similar structures within a reach and/or catchment. The study also highlights the need to correctly represent a gap in LD models, as well as the need to consider adopting a behavioural approach to the numerical modelling of such structures.

## 7. Author contributions

JW and CS conceived the study and designed the experiments. CS added the LD module into CAESAR-Lisflood codebase. JW performed the simulations, data analysis and writing of the first draft. JW, CS, DM, RT and DP equally contributed to discussing and interpreting the results and finalising the draft.

## 8. Code availability

The current version of CAESAR-Lisflood is available from https://sourceforge.net/projects/caesar-lisflood/files/ under the GNU General Public Licence version 3.0 (GPLv3).

The exact version of the model used to produce the results using in this paper are archived used Zenodo and available at https://doi.org/10.5281/zenodo.12795495, as are input data and scripts to run the model for simulations presented in this paper.

## 9. Competing interests

The corresponding author has declared that none of the authors have any competing interests.

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
