# Peer review of "Hydro-geomorphological modelling of leaky wooden dam efficacy from reach to catchment scale with CAESAR-Lisflood 1.9j"

_EGUsphere, 2024_

## Author Comment (AC1)

**Response to reviewers regarding Manuscript ID egusphere-2024-2132 entitled:**

**"Hydro-geomorphological modelling of leaky wooden dam efficacy from reach to catchment scale with CAESAR-Lisflood 1.9j" to Geoscientific Model Development.**

We would like to thank Paul Quinn and the two anonymous Reviewers for their feedback on this manuscript, as well as Andy Wickert for overviewing the submission and review process. We have responded to all Reviewer comments as detailed below.

Original text from the Topic Editor and Reviews is blue.

Responses are black, Unedited text is grey, Edited text is **bold and black**.

**Reviewer 1**

General summary

This paper presents and tests a new extension of the CAESAR-Lisflood landscape evolution model, that enables hydro-geomorphological modelling of in-stream wooden leaky barriers. The paper is interesting and well presented. It is also novel by being the first model published that considers both geomorphic and hydrological processes and the interactions between them which is of great importance for practitioners of natural flood management (NFM). The topic is also relevant to the EGU Sphere Geoscientific Model Development journal.

We thank the reviewer for their interest in the manuscript and support in acknowledging the importance of the work.

The paper has potential as a worthy contribution of a new model tool with some interesting insights into processes and responses that could have real world implications. However, the paper would benefit from some improvements. Firstly, a more up to date review and comparison with the topic of the hydro-geomorphological functioning of leaky barriers is needed. Much progress has been made recently especially with regards to modelling the hydraulic and hydrological effects of leaky barriers but these studies are not included and may help to improve the insights given in the discussion.

Our initial aim was to remain specific to the representation of leaky dams in terms of numerical modelling, however we agree that the addition of a focused review of the functioning of leaky dams would benefit the manuscript. We have added more reference to the literature including field observations, numerical modelling and flume studies as detailed below in the line-by-line comments.

Secondly, a clearer research context to help justify the paper with better stated aims are needed. For example, at the moment there is little exploration of why it is important to consider geomorphic processes and what the specific geomorphic aims of this study are.

We have added this in response to line-by-line comments as detailed below.

Thirdly, more consideration of the validity of predictions and applicability of the tool in the real world is required. The authors state that the model is heuristic, and it is unvalidated which means its reliability for making worthwhile predictions is unknown. However, at the same time

the authors advocate that the model is useful for practitioners but given the uncertainty of predictions perhaps this is not a valid viewpoint to take.

The aim of the model is to understand the relative impact of leaky dams on both the hydrology of the system and the geomorphology. As the reviewer correctly indicated in a specific comment below (referring to L477–480), there isn't the data to calibrate or validate sediment transport in the real world. Therefore, our adaptation to the CAESAR-Lisflood model in its current format is a useful addition for exploring the potential impacts of different leaky dam designs—in terms of height, gap, roughness and location—as part of a preliminary scoping study. We agree that future work is required to refine the applicability to functioning systems, however this cannot come without first addressing the data paucity surrounding leaky dams, sediment transport, and the geomorphological change they may induce. We have addressed this point throughout the line-by-line comments below.

**Specific points**

L24 State recurrence interval or annual exceedance probability of storm event to give an idea of its magnitude. Also state for what catchment size.

We have adjusted this sentence as follows:

> The findings show that simulating sediment transport increased the volume of water stored in the test reach **(channel length 160 m)** by up to an order of magnitude whilst reducing discharge by up to 31% during a storm event **(6 h, 1 in 10-year event)**.

L32-35 Low risk yes but potential for structure washout and displacement perhaps should be acknowledged.

This now reads:

> NFM is becoming increasingly popular with flood risk managers due to its multiple benefits and perceived low risk**, however due to altering the hydrological regime, there is potential for structures to become displaced and washed out (e.g., Nisbet et al., 2015). NFM is also an effective method to** engage local communities and land users in potentially reducing flood risk (Burgess-Gamble et al., 2017; Dadson et al., 2017; Newson et al., 2021).

L36 What is meant by 'river engineering'? Seems like a vague term to use.

To improve clarity, we have removed the reference to river engineering.

L46-48 More nuance and specific reference to sources that back up these claims is needed. These benefits are often perceived and have not been quantified comprehensively or assessed.

We have adapted this sentence as follows to include recent publications in support of this:

> **Reintroduction of wood to the river channel is a popular form of NFM, employed for multiple co-benefits such as habitat creation and ecological enhancements (e.g., Wohl, 2017; Ockelford et al., 2024) as well as flood peak reduction (e.g., van Leeuwen et al., 2024; Villamizar et al., 2024).**

L50-62 A more up to date and accurate reflection on the knowledge gaps and recent advancements on understanding is needed. For example, see the work of Follett and Hankin (2022) and Geertsema et al., (2020) on approaches to model the hydraulic effects of in channel large wood interventions. Also, the work of Lo et al. (2022) gives field based observations on the

geomorphic effects of leaky barriers and the work of Norbury et al. (2021) and Van Leeuwan et al. (2024) measure the hydrological effects of structures using field data. Flume based studies on hydro-geomorphic responses are also potentially useful to synthesise and compare with (e.g. Schalko et al., 2019; Muhawenimana et al., 2021).

We agree that including a summarised review of knowledge gaps and recent advancements would improve the manuscript. As such we have updated this section to provide greater understanding on the influence of large wood and LDs on geomorphology, as well as numerical and flume studies:

Despite their rapid deployment in riverine management over recent years, a key knowledge gap is how LD efficacy evolves temporally, both in response to geomorphic evolution up- and downstream of the LD, but also in response to flood sequences (Addy and Wilkinson, 2019; Grabowski et al., 2019). **The influence of large wood on river systems is well understood: wood increases fluvial complexity whilst being resistant to erosion and providing storage space for water (Gurnell et al., 2018; Wohl et al., 2019). Specifically, large wood can form pools (e.g., Abbe & Montgomery, 1996; Al-Zawaidah et al., 2021; Ravazzolo et al., 2022), increase sediment storage (e.g., Comiti et al., 2008; Wohl & Beckman, 2014), protect against or induce bank erosion (e.g., Abbe et al., 2018; Galia et al., 2024) and influence floodplain morphology (e.g., Sear et al., 2010; Wohl, 2013). Large wood is generally mobile (Wohl et al., 2023), unlike LDs that are often engineered and anchored in-situ and therefore can be functionally different with wide-ranging designs (Lashford et al., 2022; Lo et al., 2022; Quinn et al., 2022).**

Challenges in disentangling the relative impact of LDs from the influence of land use, antecedent conditions and other flood risk management interventions presently result in an unclear understanding of their influence over time. **Similar to natural wood, LDs influence the hydraulic regime through increasing roughness and thus have the potential to influence channel geomorphology. The few empirical field studies that have focussed on LDs have highlighted that LDs can reduce peak flows for the 1-year annual exceedance probability (AEP) by 10% on average, however the response can be highly variable (Norbury et al., 2021; van Leeuwen et al., 2024). The backwater rise induced by LDs is also variable and can be increased or decreased with the presence of porosity-reducing material (Muhawenimana et al., 2023). Furthermore, the ability of a LD to store water, or sediment, can be dependent on the distance between the riverbed and the bottom of the LD, with gaps >0.3 m unable to store sediment in the Yorkshire Dales, UK (Lo et al., 2022), while increased wood volume also amplifies scour (Schalko et al., 2019). Laboratory experiments have shown that representing LDs as non-porous structures increases drag and flow area (Muhawenimana et al., 2021), and therefore it is important to account for porosity of the structures in numerical simulations. Yet often porosity is not considered in numerical simulations due to representing these complex structures in reduced-complexity models.**

Recent works have focused on integrating LDs into 1D and 2D models at different spatial scales (Hill et al., 2023), most commonly representing the interventions as localised roughness adjustments (Pinto et al., 2019; **Geertseema et al., 2020**), geometry adjustments (Pearson, 2020; Walsh et al., 2020), or a combination of the two (Dixon et al., 2016; Senior et al., 2022). LDs have also been represented in hydraulic models, through stage-discharge relationships realising LDs (and other RAFs) as weirs or culverts (Thomas and Nisbet, 2012; Metcalfe et al., 2017; Keys et al., 2018; Hankin et al., 2019; Pinto et al., 2019; Hankin et al., 2020; Leakey et al., 2020; Pearson, 2020; **Follett and Hankin, 2022**). A comprehensive review of the large wood

numerical modelling literature focused on artificially placed wood can be found in Addy and Wilkinson (2019).

L64-65  Clear statements on potential geomorphic processes, feedbacks and importance are needed.  At the moment the importance of considering geomorphic processes in models isn't coming through.  For example, the aforementioned studies have given observations on the patterns of erosion and deposition in relation to structures that could have hydraulic effects.

Following on from the previous paragraphs that now more clearly state this, we have incorporated the following:

**LDs and large wood clearly can alter local morphology, which in turn can alter hydraulic response through feedback cycles of erosion and deposition (Lo et al., 2021).**

L81-82 First part of this statement is not true.  See references made above on progress made on representing the leakiness and lower gap effects of leaky barriers.

We have adjusted this sentence as follows to increase clarity around this point:

As such, no work currently exists that incorporates both the inherent 'leakiness' of LDs and the ability to simulate a lower gap **coupled with** sediment transport **to evaluate** geomorphic evolution **within a numerical model**.

L82 What is meant by a 'prototype real world location'?

For clarity we have removed the word "prototype".

L84  A new paragraph stating a clearer and more elaborated list of aims and, or hypotheses is needed.  This would help to give the paper more structure and purpose.

We have added the following text in support of this:

**The aims of this paper were to explore the relative behavioural impact of a simple LD on sediment transport processes and subsequent influences on discharge and water storage through a small reach. To do this, we first introduce new functionality for CAESAR-Lisflood that can represent LDs through the restriction of flow. Second, we evaluate the sensitivity of the model to DEM resolution, and third, assess the impact of LD roughness and gap size on geomorphology and water storage. Finally, we present the implications of numerically simulating LDs coupled with sediment transport processes to inform future modelling studies.**

L159-160  'Upwinding' and 'upwind' are strange terms to use.  Consider rewording?

The upwind scheme is the correct term for calculating the downstream flow field based on cells upstream, however for clarity we have updated this to:

CL employs a first order **upwind scheme (Coulthard et al., 2013).**

L190 This approach of scaling n seems sensible but perhaps a caveat is needed here given that it isn't based on an empirical relationship as it stands.

We have added the below caveat immediately following the introduction of BR:

**BR captures increasing LD roughness with increasing stage to reflect increasing complexity as a greater vertical area of the channel is obstructed. Further empirical data is required to assess this assumption.**

 More details on the prototype reach is needed in the paper rather than citing the thesis.  The channel slope used seems quite low for a headwater stream where leaky barriers are typically used.  Would it be worth testing the model over a range of slopes to see the effect?  Why were different DEM resolutions tested?  Certainly important but little context or purpose on this is given.

We have added more details to this section as outlined below:

> The model domain was 160 m long and 100 m wide **and represents a second-order stream**. The DEM had the same average slope as a prototype site (0.01 m m$^{-1}$; Wolstenholme, 2023) **where LDs were installed in 2019** and was created by linear interpolation between the high and low survey points in the reach captured with a Topcon OS-103 Total Station (TS).

We explored the impact of grid resolution to understand whether any bias was introduced in the model outputs, and to ensure that potential caveats as a result of resolution were highlighted in this development and technical paper.  We agree that it would be interested to explore the impact of slopes on the effect, however this is beyond the current scope of the manuscript.

L221 What is meant by sediment types?

Sediment types referred to the different grain size bins used in CAESAR-Lisflood, however upon reflection this was confusing. Therefore, we have adjusted the sentence as follows:

> This ensured that **the sediment was** distributed throughout the catchment in equilibrium with the topography.

L238 Again like testing a range of slopes, it would be interesting to see the effects of a range of different flow events and perhaps would provide more insight into hydrogeomorphic effects of leaky barriers than testing a range of different DEM cell sizes.

As mentioned above, we agree that this would be interesting, however is beyond the scope of the development and technical paper where we focus more on the implementation of leaky dams in CAESAR-Lisflood.

L246 Why was an nmax value of 0.16 used?

This was chosen as a conservative estimate of the leaky dam roughness based on empirical studies (e.g., Curran and Wohl, 2003; Dixon et al., 2016 and Addy and Wilkinson, 2019). Second, we explore the relative impact of nmax in section 4.3 from 0.12–0.2, and for the purposes of not biasing the impact of grid resolution or gap size with changing multiple parameters in parallel, we chose the central value (0.16) of this range.

To provide this justification in the text, we have made the following adjustments:

> ...dams was set to 0.16 **(chosen as a conservative estimate of LD roughness after Curran and Wohl, 2003; Dixon et al., 2016; Addy and Wilkinson, 2019 )**.

Figure 6 To make it clear, mark on erosion and deposition labels.  I.e. negative values show deposition and positive values show erosion which may be counter-intuitive at first glance.  Similar remarks can be made for Figure 7 in relation to changes in water storage.

Negative values represent erosion and positive, deposition. We inverted the standard output from CAESAR-Lisflood to ensure that the elevation change was not misconstrued. We have, however, added text to that effect to Figure 6 to ensure that there is no confusion.

Figure 7 only shows water storage therefore we have not altered this figure.

L343-344 The finding that increasing LD gap size resulted in less sediment being lost seems counter-intuitive.

Less sediment was lost out of the model domain. There was less erosion upstream and downstream of the leaky dams when gap size was increased (e.g., Figure 9) as the river had less energy to erode the bed when compared to a smaller gap size.

L455-459 Perhaps a more important point is how valid is this model. Whilst there may be some value in using it in an heuristic manner for scenario testing, validation using empirical observations of changes in discharge and morphology would strengthen the value of this model and give practitioners more confidence in using it. This is a knowledge gap that should be stated. The word 'behaviouralist' seems a strange one to use.

We agree that to add more value the discussion would benefit from being more closely tied in with existing leaky dam literature. Therefore, to address this point we have included the following text at the end of section 5.2.:

> Care must be taken when using the LD module for CL as the right results, such as elevation change and $Q$ reduction, may be overestimated when using coarser grid resolutions. **The scenarios simulated herein align well with the relative influence of LD from both field and laboratory observations including the formation of downstream pools (Lo et al., 2021; Lo et al., 2022; Muhawenimana et al., 2023), and the potential for sediment storage upstream (Comiti et al., 2008). Future work should focus on calibrating and developing this tool as a flexible and rapidly deployable option for LD simulations in CL, that should currently be used heuristically to mitigate the need for calibration. The LD module for CL can therefore be best used to understand the relative impact of LDs in larger, complex catchment to identify their individual impact on FRM.**

L477-480 Yes this is a fair statement but there simply isn't usually the data to calibrate or validate sediment transport in the real world unfortunately and processes are notoriously difficult to predict for different hydrological events. In contrast, observations of channel hydraulics and hydrology are more available o accessible to enable model assessment.

We are glad that the reviewer agrees with this statement, and that the lack of data to calibrate/validate sediment transport highlights the need for the heuristic approach adopted and subsequent analysis.

L491 Could the model be used to assess other types of NFM structure or in channel wood? Another aspect deserving further study in future?

Absolutely. We have added the following to the previous paragraph to highlight this:

> **Additionally other types of structures with porosity can be evaluated with the model, if it can be defined through its height, gap (or lack of) and roughness, for example natural wood, bridges and bunds, and presents opportunities for further study.**

Associated files with the paper. Appears that the model has been uploaded for both weblinks on Zenodo. Could not see the datasets.

All data used in the study and the model are available on Zenodo, in the .7z file.

**Typo errors**

 This sentence does not make sense.

Apologies, this has been corrected to:

> **Sediment transport becomes increasingly important when unintended geomorphic adjustment to 'hard engineered' structures reduces the efficacy of structures, potentially increasing flooding downstream (Hesselink et al., 2003; Pinter et al., 2006; Hudson et al., 2008; Benito and Hudson, 2010).**

**References**

Follett, E. and Hankin, B., 2022. Investigation of effect of logjam series for varying channel and barrier physical properties using a sparse input data 1D network model. Environmental Modelling & Software, 158: 105543.

Geertsema, T.J., Torfs, P.J., Eekhout, J.P., Teuling, A.J. and Hoitink, A.J., 2020. Wood-induced backwater effects in lowland streams. River Research and Applications, 36(7): 1171-1182.

Lo, H.W., van Leeuwen, Z., Klaar, M., Woulds, C. and Smith, M., 2022. Geomorphic effects of natural flood management woody dams in upland streams. River Research and Applications, 38(10): 1787-1802.

Muhawenimana, V., Wilson, C.A., Nefjodova, J. and Cable, J., 2021. Flood attenuation hydraulics of channel-spanning leaky barriers. Journal of Hydrology, 596: 125731.

Norbury, M., Phillips, H., Macdonald, N., Brown, D., Boothroyd, R., Wilson, C., Quinn, P. and Shaw, D., 2021. Quantifying the hydrological implications of pre- and post-installation willowed engineered log jams in the Pennine Uplands, NW England. Journal of Hydrology, 603: 126855.

Schalko, I., Lageder, C., Schmocker, L., Weitbrecht, V. and Boes, R.M., 2019. Laboratory Flume Experiments on the Formation of Spanwise Large Wood Accumulations: Part II—Effect on local scour. Water Resources Research, 55(6): 4871-4885.

van Leeuwen, Z.R., Klaar, M.J., Smith, M.W. and Brown, L.E., 2024. Quantifying the natural flood management potential of leaky dams in upland catchments, Part II: Leaky dam impacts on flood peak magnitude. Journal of Hydrology, 628: 130449.

**Reviewer 2 (Paul Quinn)**

I would first like to congratulate Reviewer 1 for a very detailed analysis of the paper and that I fully agree with these recommendations. So, I will concentrate on general matters.

I think more papers are needed including more general reviews such as Lashford et al., 2022 and Quinn et al.,, 2022. As they pose important design concerns. Which is what my main discussion point is.

Thank you for your comments. We have added references around the differences of designs in the introduction (now lines 59–61):

> **Large wood is generally mobile (Wohl et al., 2023), unlike LDs that are often engineered and anchored in-situ and therefore can be functionally different with wide-ranging designs (Lashford et al., 2022; Lo et al., 2022; Quinn et al., 2022).**

We also have made more specific reference to the ability of the model to simulate different designs (section 2.2):

> The approach developed herein represents the leakiness of a LD, its water depth-dependent impact on the water column, and changing efficacy due to implicit geomorphic changes. **There are multiple different designs for LDs, from those that are more natural and better emulate natural large wood, to those that are more engineered, with slots for water to pass through (Lashford et al., 2022; Quinn et al., 2022). The module presented herein allows the** simulation of gaps below LDs, a common design feature, in a way that can alter due to erosion or deposition.

Overall, a very good paper, it is clear and it covers the scope (as stated) very well. The issue of Manning n having an impact on flow is the most important part of this paper. Perhaps plant some vegetation on the banks?

Thank you for these comments and for supporting our work. We agree that the influence of Manning's n has a clear impact on flow regulation. Currently planting vegetation on the banks is beyond the scope of this study, but we agree that future work should focus on combining leaky wooden dams with bank vegetation in numerical models.

The most concerning issue, that I think can be easily solved, is the assumption that leaky dams with a rectangular lower slot is the only design of leaky dam of concern. There are many leaky dam designs. Designs that allow a dynamic alteration of discharge with flow magnitude/depth being the obvious example. What if the leaky dam has a vertical slot or multiple pipes at multiple depths? So, you must make it clear that you are restricting your analysis to just this design and it still tell us interesting things.

The method chosen to increase Manning's n allows for different designs to be simulated, through using n as a proxy for complexity (i.e., a higher n = more flow resistance). It is possible in the model to set a gap size of 0 (so that the base of the leaky dam is in contact with the riverbed) and define n to reflect the resistance that the leaky dam may impose on flow.

We agree that the method does not allow for vertical slots or multiple pipes at multiple depths and have reflected this by including the text outlined in a previous comment. By adjusting n in the model, however, the user can iteratively tweak the response of the installed structure to reflect their use case (i.e., a more complex structure would have a higher roughness).

Your choice of storm and flows need clarification. O.1 RI rain storm or flow RI?

The recurrence interval is reflective of the rainfall as detailed in line 207 (now line 236–237):

> Sensitivity analysis was performed using a single rainfall input—a six-hour, 10% AEP **rainfall** event...

Where on the network is your reach? What is the catchment area entering the top end of the reach? This is important. You have said that you are testing sensitivity, BUT you have not tested sensitivity to changing inflow. LDs are drowned out quite quickly so if the inflow increases the effectiveness goes down quickly. Or are you stating another design restraint of your study that you assume LDs are only at the upper most part of headwater e.g. less than 1km2? In reality LDs can be effective at larger scales if designed correctly. So be clear.

The reach drains an upstream area of 2 km$^2$ and is on a second-order stream (now reflected in section 3).

We chose to keep inflow constant to explore the effects of the LDs themselves and DEM resolution. We agree that this would make an interesting follow-up study coupled with the influence of LDs on different parts of the network.

If you have assumed no storage is created being the LD, then you have missed the key design concept that LDs work best if they can engage larger storage volumes. Being constrained within a channel is the worst design.

So just broaden your discussion and conclusion.

We have not assumed that there is no storage created by the LD, rather that herein there is only storage within the channel. We have implemented the LD within a single channel to reduce complexity in the first instance, however the functionality is present within the model for users to expand the LDs across the floodplain.

**Reviewer 3**

The Landscape Evolution Model CAESAR Lisflood 1.9j is employed to simulate leaky wooden dams at reach scale. A roughness-based approach is followed, and the simulation of sediment transport is also included, which represent a significant improvement with respect to current simulation practice.

A theoretical case study is proposed, and the Author systematically analyse the results by combining the variation of several parameters, from cell size, to roughness, to lower gap size, observing the variation of hydraulic and geomorphological results. One LD is represented in a straight channel.

The contribution is certainly interesting, especially for its effort of understanding the dynamic behaviours of LD.

We thank the Reviewer for their interest and support in our work.

However, the approach is strongly theoretical, with no connection to field data, making it an interesting attempt, but still hardly applicable. At least, this limitation should be clearly discussed.

Following comments from the other Reviewers, we have included a more robust and recent overview of the literature in the introduction. The aim of the paper was to provide a behavioural/relative approach to simulating the impact of LDs on geomorphology and derive insights from the influence on local morphology and discharge outputs, to inform decisions on the potential impacts on different LD parameters.

In addition, some inaccuracies are present in the paper, like the absence of the results for certain gap size (not clearly justified), the absence of figures related to the case study domain, and a very synthetized presentation of the results, which seems under "discussion" form rather than result analysis. For example, why not to show the actual discharge hydrographs, rather than showing immediately a discharge variation?

Gap sizes 0.3 and 0.4 m had no impact as the stage was too low to engage with the leaky dam, therefore we elected to not include these figures in the text. This has been reflected throughout the manuscript now in sections 4.4.1–4.4.3.

We elected not to display the actual discharge hydrographs (with the exception of the input discharge that can be seen throughout) as actual variations were difficult to see. Instead, we normalised the differences between the baseline simulation (i.e., no LDs) and the experiments to more clearly present the relative influence of the structure and the various parameters.

The methodology relies on the application of an evolution model, and, as said, the absence of a clear calibration is the main limitation of the selected approach.

We have addressed this throughout with increased reference to the literature as detailed below, in addition to the response to other Reviewers.

Despite having employed different model resolution, the effect of the resolution is not discussed, possibly leading to results inaccuracies. For example, while erosion and sediment areas along the channel are similar, the effect on the variation of the discharge is different for different resolution. This aspect is only addressed in the final discussion, but I think it should be analysed with higher precision also in the results.

Thank you for this comment. We have addressed this in the line-by-line comments below by going into further detail as requested.

Please find below my detailed comments:

Lines 133-135: "This method is straightforward to apply within a model domain, as specific cells can be identified to place the LD in combination with other roughness variables such as in-channel or floodplain boundary roughness". How does the model identify LD cells with respect to riverbed or floodplain cells? Is the cell dimension modified for rover or floodplain?

The riverbed or floodplain cells can be separated through the user-defined "m" value or spatially-variable Manning's n value, however at present, CAESAR-Lisflood does not have the capability to transfer these values and be reflective of evolving morphodynamics. Often a global Manning's n value is applied to mitigate this issue (the method that we adopted), yet LDs can be placed anywhere in the model domain and increase the roughness of the chosen cell based on the blockage ratio as defined in the manuscript.

Figure 2: the figure shows a square with one LD, and either vertical (b) or horizontal (c) flow direction. However, I guess that the model allows also horizontal + vertical flow, in case of transversal connectivity (flooding the areas upstream of the LD). I think that the figure should highlight the possibility of such a behaviour. In addition, connected to the previous comment, the cell edge that represents the LD is representative of the physical dimension of the LD?

Flow in CAESAR-Lisflood can only enter a cell from the four cardinal directions (D4 flow algorithm) and flow diagonally across cells is not possible and therefore we have not edited this figure. Regarding the second question, the minimum width of the LD is governed by the DEM grid resolution. Multiple LD neighbouring cells can create a wider LD as seen for $DEM_{T1}$ and $DEM_{T2}$ where the LD spans 4 m.

Line 190-191: what do the Authors mean with "If the LD is overtopped, there is less of the LD in contact with the water column, therefore $n$ is reduced."? If the LD is overtopped, the water level is higher than $h_{top}$. So, the entire LD will be in contact with the water column, which however will have additional area not in contact with LD, thus reducing the $BR$. Is this the reason why the roughness coefficient is reduced? Please adjust the sentence.

Thank you for this comment, we have updated the sentence accordingly as we agree it was unclear:

> **If the LD is overtopped, the blockage ratio reduces because the cross-sectional area of flow increases while the cross-sectional area of the LD remains constant. Therefore the relative cross-sectional roughness reduces and thus *n* is reduced.**

Lines 190-192: These sentences appear reasonable but highly qualitative. Was any calibration performed to justify the variation of that roughness behaviour?

No specific calibration was performed, however multiple authors (as shown throughout the Introduction) highlight the importance of the stage relationship with LDs, and most (if not all) models that use a roughness-based approach for LD representation instantly increase roughness. The approach developed herein was designed to reduce unlikely scenarios such as instantly increasing roughness. We have additionally included the following text, following a comment from Reviewer #1:

> **BR captures increasing LD roughness with increasing stage to reflect increasing complexity as a greater vertical area of the channel is obstructed. Further empirical data is required to assess this assumption.**

Paragraph 3.1: A figure representing the DEM and the different resolution would help the reader in figuring out tests configuration.

We have chosen not to produce this figure as there is little visual difference between the three DEMs due to the simple nature of the test reach.

Figure 3: Please also include the b-axis sketch (or a description in the text), to ensure the proper interpretation from readers that may be not confident with the grain-size analysis.

The Wolman pebble count is a standard and well-known geographical surveying method therefore we do not believe that this is required. However, we have added the following text for clarification for those less familiar:

> The b **(intermediate)** axis of >400 randomly…

Paragraph 3.3: the time-step of the simulations is not reported.

We have updated the text to include the timestep of the simulations:

> …120-hour period of baseflow **with a 60 second timestep.**

Figure 4: To ease the readability of the figure, why not to use the SI units fo measurements also for time (seconds or hours, instead of minutes x $10^4$)? I also believe that showing the output discharge hydrograph, for at least one of the tests, would help in clarifying the discussion,

Thank you for this suggestion, we have implemented the SI units throughout these figures. Regarding the output discharge hydrograph, as mentioned in an earlier comment, we chose specifically not to do this in favour of normalised differences to better highlight the effect of the LD.

Lines 271-273: the Authors say: "For all sediment transport enabled experiments, the falling limb of the storm and remainder of the simulation time shows up to 25% deviation from the baseline scenario due to $Q$ becoming out of phase with the baseline.". In Figure 4, C and D, a variation of the discharge is observed, the values od ΔQ being positive or negative depending on the cells size. Positive values reach 25% for $DEM_{T4}$ and reached -10% for $DEM_{T1}$. Which of the situations is reflected in the Authors' sentence? I think that here you should further discuss the effect of the cells size in providing attenuation/incrementation of the simulated discharges.

This sentence refers to all sediment transport enabled experiments, as stated. The variation was ±25% (we have amended this slightly by adding the ± symbol) as shown in the Figure (now Figure 5). We have also added further descriptive text around the sentence to highlight the influence of the LD and sediment transport more effectively as follows:

> For all sediment transport enabled experiments, the falling limb of the storm and remainder of the simulation time shows up to **±**25% deviation from the baseline

scenario due to $Q$ becoming out of phase with the baseline. **DEM$_{T1}$ had an overall reduction in $Q$ (maximum = –10%) following the peak of the storm compared to coarser grid resolutions DEM$_{T2}$ and DEM$_{T4}$ that increased $Q$ (maximum +12% and +25%, respectively). Where $h_{gap} = 0.2$, although the influence of the LD had little impact on the peak, there was much larger disruption to the discharge on the falling limb. These disruptions represent a deviation in $Q$ of up to 0.014 m$^3$s$^{-1}$ as a result of sediment transport and the presence of the LD.**

Line 280: $Q_s$ was not introduced before. Please, define it clearly.

We have defined Q$_s$ (sediment flux).

Line 310: The Authros wrote: "the second most cell downstream of the LD". I suggest rephrasing as follows: "the second cell most downstream of the LD".

Done.

Figure 6: The amount of change appear strongly dependant on the cells size. Please add a comment about that.

The relative elevation change increases as grid resolution increase, however the overall pattern remains constant regardless of grid resolution. To emphasise this, we have updated the text in section 4.3. as follows:

> All simulations **regardless of grid resolution follow the same pattern of elevation change. There was** increased deposition 60–100 m downstream (average +0.12 m, maximum +0.25 m).

Lines 318-319: "a 2 m DEM with a two-pixel wide channel (4 m)". Do the Authors refer to DEM$_{T2}$? If yes, please use the same acronym, to be consistent within the paper. If not, please explain better.

Yes, we have updated this.

Paragraph 4.3.1: Is only no-gap simulation discussed? If yes, please include general comments also for other gaps. In addition, why is Figure 7 not showing 0.3 and 0.4 m gaps? They should be discussed too, since they are mentioned.

We have updated the paragraph as follows:

> Sediment transport enabled simulations showed at least an order of magnitude greater water storage than the hydraulic equivalent. Where there was no LD gap, water was instantly stored upon LD installation until the onset of the storm where rougher LDs were found to store the greatest volume of water. **Larger gap sizes did not store water upon LD installation, rather when stage reached the base of the LD. Where the gap size was 0.3 m or greater, there was no difference in water storage and therefore the LD did not engage with the river. For all remaining scenarios,** water storage **was** greatest during the peak of the storm**, with increasing gap size resulting in diminished storage during the peak for both hydraulic and sediment simulations. There was no subsequent water storage for hydraulic simulations except where there was a 0 m gap. In contrast** when simulating sediment transport, the system **stored** 0.95–2.2 m$^3$ of water when compared to the baseline experiment, as shown in Figure 7**, with diminishing effectiveness when gap size was increased.**

 the Authors say: "Sediment transport was found not to be influenced by the installation of the LD". However, in Figure 8 a variation curve in the sediment yield is shown. Please consider modifying this sentence. Probably the variation is not significant nor clear, but the sediment yield appears to be at least slightly different from the baseline one. In addition, discuss also the results for 0.3 and 0.4 m gaps.

The installation of the LD (at 20 hours) had no impact on sediment transport as there is no variation in the sediment discharge therefore, we have not altered the Figure. Regarding the 0.3 and 0.4 m gaps, these are not discussed as the flow did not engage with the LD, as such there was no difference from the baseline scenario. We have updated the text to reflect this (see comment below).

Figure 8: why are subfigures A', B' and C' displayed? They are not commented in the text.

These have now been integrated into the text as described in the following comment.

Lines 342-343: Please discuss why with a larger gap sediment storage is observed.

We have added the following text to discuss this:

> Sediment transport was found not to be influenced by the installation of the LD **at 20 hours**, regardless of gap size**, however only when the gap was 0.2 m or less, did the LD influence $Q_s$ during the storm** (see Figure 8). **Prior to the peak of the storm, $Q_s$ was reduced by <0.03 m³ when the LD engaged with the flow** compared to the baseline scenario **(see Figure 8A'–C')**, but there was little variability prior to the peak (average standard deviation: 0.008). Following the peak of the storm, the influence of roughness variability was more pronounced **with an increase in $Q_s$** ($\Sigma\Delta Q_s$ 0.25–0.8 m³), however there **was** no clear trend between the volume of sediment exiting the model domain (Figure 8) and the **chosen roughness value**. However, increasing LD gap size did result in less sediment being lost out of the domain, **likely due to there being a lower energy gradient induced by the LD**.

Figure 9: Also here, 0.3 and 0.4 m gaps are not shown.

The text has been updated near the top of paragraph 4.3.3. with:

> **When the LD gap was 0.3 m or greater, there was no influence on bed morphology.**

Additionally, alongside Figures 7 & 8, the following text has been added to the caption:

> **0.3 m and 0.4 m gap simulations were omitted due to their having no impact.**

Lines 374-376: the Authros suggest that: "CL can therefore be used to aid identification of the ideal locations for LDs throughout a given reach to achieve the desired behavioural response, such as increased flow attenuation or enhanced geomorphic diversity." However, the approach here proposed only shows the potentiality of the model. In a real-case scenario, will the peak reduction, the erosion and deposition occur according to the model predictions. In my opinion, the approach is really interesting but, before being employed for NFM design, it needs to be calibrated and validate on different case-studies. This does not mean reduce the importance of the contribution but helps in identifying its limitations and stress the requirements of future efforts.

Thank you for this comment. We agree with your statement on calibration, however the objective of this work was to explore the relative impact in a test-reach. Therefore, we have updated the sentence to be more reflective of work required for use in a real reach:

CL **could** therefore be used to **explore the relative influence of LDs and their parameters** throughout a given reach to **indicate a** behavioural response, such as increased flow attenuation or enhanced geomorphic diversity **following calibration**.

Lines 388-389: "In addition, as CL utilises a regular raster grid, the required resolution can be adapted based on user requirements as well as computational resource availability".

Discussion: when discussing about computational burden, please consider including tables with the following data: cell resolution, number of cells, computational time. This will help the paper to be clearer on the effect of grid size on computational burden.

We have added a new section: 4.1. Computational expense to introduce the computational expense of the LD module with respect to DEM resolution and chosen parameters, including a new figure (now Figure 4) that visualises the expense:

> **Figure 4 shows the number of iterations required to complete the simulations for the performed experiments. Iterations are a useful proxy for model efficiency, independent of the computer used to perform the simulations. They show that for decreasing grid resolution, fewer iterations are required across all experiments. Simulation of the LD with only the hydraulic model enabled results in little increase in computational expense (averaged standard deviation = 20,259) whereas enabling sediment transport drastically increased model iterations often by over 200% across all DEM resolutions.**

Line 469: I think it should be "reduced" and no "reduce"; or "can reduce"?

We have updated this as follows:

> **Sediment transport becomes increasingly important when unintended geomorphic adjustment to 'hard engineered' structures reduces the efficacy of structures, potentially increasing flooding downstream (Hesselink et al., 2003; Pinter et al., 2006; Hudson et al., 2008; Benito and Hudson, 2010).**

Line 488: "which could further the understanding". A verb is probably missing.

We have updated this as follows:

> ...given timestep, **furthering** the understanding...

---

## Referee Report (RR1)

**Response to reviewers regarding Manuscript ID egusphere-2024-2132 entitled: "Hydro-geomorphological modelling of leaky wooden dam efficacy from reach to catchment scale with CAESAR-Lisflood 1.9j" to Geoscientific Model Development.**

We would like to thank Paul Quinn and the two anonymous Reviewers for their feedback on this manuscript, as well as Andy Wickert for overviewing the submission and review process. We have responded to all Reviewer comments as detailed below.

Original text from the Topic Editor and Reviews is blue.

Responses are black, Unedited text is grey, Edited text is **bold and black**.

Reviewer 1 responses to first paper revision and author comments given below in red text.

Reviewer 1 General summary

This paper presents and tests a new extension of the CAESAR-Lisflood landscape evolution model, that enables hydro-geomorphological modelling of in-stream wooden leaky barriers. The paper is interesting and well presented. It is also novel by being the first model published that considers both geomorphic and hydrological processes and the interactions between them which is of great importance for practitioners of natural flood management (NFM). The topic is also relevant to the EGU Sphere Geoscientific Model Development journal.

We thank the reviewer for their interest in the manuscript and support in acknowledging the importance of the work.

Ok.

The paper has potential as a worthy contribution of a new model tool with some interesting insights into processes and responses that could have real world implications. However, the paper would benefit from some improvements. Firstly, a more up to date review and comparison with the topic of the hydro-geomorphological functioning of leaky barriers is needed. Much progress has been made recently especially with regards to modelling the hydraulic and hydrological effects of leaky barriers but these studies are not included and may help to improve the insights given in the discussion.

Our initial aim was to remain specific to the representation of leaky dams in terms of numerical modelling, however we agree that the addition of a focused review of the functioning of leaky dams would benefit the manuscript. We have added more reference to the literature including field observations, numerical modelling and flume studies as detailed below in the line-by-line comments.

This has now been improved but see specific clarifications required below.

Secondly, a clearer research context to help justify the paper with better stated aims are needed. For example, at the moment there is little exploration of why it is important to consider geomorphic processes and what the specific geomorphic aims of this study are.

We have added this in response to line-by-line comments as detailed below.

Now improved.

Thirdly, more consideration of the validity of predictions and applicability of the tool in the real world is required. The authors state that the model is heuristic, and it is unvalidated which means its reliability for making worthwhile predictions is unknown. However, at the same time the authors advocate that the model is useful for practitioners but given the uncertainty of predictions perhaps this is not a valid viewpoint to take.

The aim of the model is to understand the relative impact of leaky dams on both the hydrology of the system and the geomorphology. As the reviewer correctly indicated in a specific comment below (referring to L477–480), there isn't the data to calibrate or validate sediment transport in the real world. Therefore, our adaptation to the CAESAR-Lisflood model in its current format is a useful addition for exploring the potential impacts of different leaky dam designs—in terms of height, gap, roughness and location—as part of a preliminary scoping study. We agree that future work is required to refine the applicability to functioning systems, however this cannot come without first addressing the data paucity surrounding leaky dams, sediment transport, and the geomorphological change they may induce. We have addressed this point throughout the line-by-line comments below.

Again, see comments below regarding this.

**Specific points**

L24 State recurrence interval or annual exceedance probability of storm event to give an idea of its magnitude. Also state for what catchment size.

We have adjusted this sentence as follows:

The findings show that simulating sediment transport increased the volume of water stored in the test reach **(channel length 160 m)** by up to an order of magnitude whilst reducing discharge by up to 31% during a storm event **(6 h, 1 in 10-year event)**.

Still need detail on catchment size, this important context to present.

L32-35 Low risk yes but potential for structure washout and displacement perhaps should be acknowledged.

This now reads:

NFM is becoming increasingly popular with flood risk managers due to its multiple benefits and perceived low risk**, however due to altering the hydrological regime, there is potential for structures to become displaced and washed out (e.g., Nisbet et al., 2015). NFM is also an effective method to** engage local communities and land users in potentially reducing flood risk (Burgess-Gamble et al., 2017; Dadson et al., 2017; Newson et al., 2021).

Ok this looks better.

L36 What is meant by 'river engineering'? Seems like a vague term to use.

To improve clarity, we have removed the reference to river engineering.

Ok.

L46-48 More nuance and specific reference to sources that back up these claims is needed. These benefits are often perceived and have not been quantified comprehensively or assessed.

We have adapted this sentence as follows to include recent publications in support of this:

**Reintroduction of wood to the river channel is a popular form of NFM, employed for multiple co-benefits such as habitat creation and ecological enhancements (e.g., Wohl, 2017; Ockelford et al., 2024) as well as flood peak reduction (e.g., van Leeuwen et al., 2024; Villamizar et al., 2024).**

But there are still deficiencies of understanding and assessment of LDs specifically which may differ to other types of large wood addition (e.g. bank attached or medial structures used to improve habitat) or naturally occurring large wood structures. For example, the number of studies that quantify empirically the flood peak mitigation of these structures is limited. Also, the study of Villamizar is based on a model whereas the van Leeuwen study is empirically based, this distinction should be made.

L50-62 A more up to date and accurate reflection on the knowledge gaps and recent advancements on understanding is needed. For example, see the work of Follett and Hankin (2022) and Geertsema et al., (2020) on approaches to model the hydraulic effects of in channel large wood interventions. Also, the work of Lo et al. (2022) gives field based observations on the geomorphic effects of leaky barriers and the work of Norbury et al. (2021) and Van Leeuwan et al. (2024) measure the hydrological effects of structures using field data. Flume based studies on hydro-geomorphic responses are also potentially useful to synthesise and compare with (e.g. Schalko et al., 2019; Muhawenimana et al., 2021).

We agree that including a summarised review of knowledge gaps and recent advancements would improve the manuscript. As such we have updated this section to provide greater understanding on the influence of large wood and LDs on geomorphology, as well as numerical and flume studies:

Despite their rapid deployment in riverine management over recent years, a key knowledge gap is how LD efficacy evolves temporally, both in response to geomorphic evolution up- and downstream of the LD, but also in response to flood sequences (Addy and Wilkinson, 2019; Grabowski et al., 2019). **The influence of large wood on river systems is well understood (1): wood increases fluvial complexity whilst being resistant to erosion and providing storage space for water (2) (Gurnell et al., 2018; Wohl et al., 2019). Specifically, large wood can form pools (e.g., Abbe & Montgomery, 1996; Al-Zawaidah et al., 2021; Ravazzolo et al., 2022), increase sediment storage (e.g., Comiti et al., 2008; Wohl & Beckman, 2014), protect against or induce bank erosion (e.g., Abbe et al., 2018; Galia et al., 2024) and influence floodplain morphology (e.g., Sear et al., 2010; Wohl, 2013). Large wood is generally mobile (Wohl et al., 2023), unlike LDs that are often engineered and anchored in-situ and therefore can be functionally different with wide-ranging designs (Lashford et al., 2022; Lo et al., 2022; Quinn et al., 2022).**

Challenges in disentangling the relative impact of LDs from the influence of land use, antecedent conditions and other flood risk management interventions presently result in an unclear understanding of their influence over time. **Similar to natural wood, LDs influence the hydraulic regime through increasing roughness and thus have the potential to influence channel geomorphology. The few empirical field studies that have focussed on LDs have**

**highlighted that LDs can reduce peak flows for the 1-year annual exceedance probability (AEP) by 10% on average (3), however the response can be highly variable (Norbury et al., 2021; van Leeuwen et al., 2024). The backwater rise induced by LDs is also variable and can be increased or decreased with the presence of porosity-reducing material (Muhawenimana et al., 2023). Furthermore, the ability of a LD to store water, or sediment, can be dependent on the distance between the riverbed and the bottom of the LD, with gaps >0.3 m unable to store sediment in the Yorkshire Dales, UK (Lo et al., 2022), while increased wood volume also amplifies scour (Schalko et al., 2019). Laboratory experiments have shown that representing LDs as non-porous structures increases drag and flow area (Muhawenimana et al., 2021), and therefore it is important to account for porosity of the structures in numerical simulations. Yet often porosity is not considered in numerical simulations due to representing these complex structures in reduced-complexity models.**

Recent works have focused on integrating LDs into 1D and 2D models at different spatial scales (Hill et al., 2023), most commonly representing the interventions as localised roughness adjustments (Pinto et al., 2019; **Geertseema et al., 2020**), geometry adjustments (Pearson, 2020; Walsh et al., 2020), or a combination of the two (Dixon et al., 2016; Senior et al., 2022) **(4)**. LDs have also been represented in hydraulic models, through stage-discharge relationships realising LDs (and other RAFs) as weirs or culverts (Thomas and Nisbet, 2012; Metcalfe et al., 2017; Keys et al., 2018; Hankin et al., 2019; Pinto et al., 2019; Hankin et al., 2020; Leakey et al., 2020; Pearson, 2020; **Follett and Hankin, 2022**). A comprehensive review of the large wood numerical modelling literature focused on artificially placed wood can be found in Addy and Wilkinson (2019).

Please see numbers inserted into text above:

(1) Assume this is a reference to naturally formed and occurring large wood?
(2) Do you mean large wood has resistance to being transported? Better state resistance to displacement instead to be clearer? The statement that large wood provides storage space for water seems a bit strange as by adding wood to a stream the volume of in channel storage would be reduced. Do you mean the ability of large wood to reconnect floodplains and create out of bank storage?
(3) Did both studies (Norbury and van Leeuwan) show the same reductions in peak flow for the same size of flow? Be clear.
(4) Are these hydraulic or hydrological models?

L64-65 Clear statements on potential geomorphic processes, feedbacks and importance are needed. At the moment the importance of considering geomorphic processes in models isn't coming through. For example, the aforementioned studies have given observations on the patterns of erosion and deposition in relation to structures that could have hydraulic effects.

Following on from the previous paragraphs that now more clearly state this, we have incorporated the following:

**LDs and large wood clearly can alter local morphology, which in turn can alter hydraulic response through feedback cycles of erosion and deposition (Lo et al., 2021).**

Ok.

L81-82 First part of this statement is not true. See references made above on progress made on representing the leakiness and lower gap effects of leaky barriers.

We have adjusted this sentence as follows to increase clarity around this point:

As such, no work currently exists that incorporates both the inherent 'leakiness' of LDs and the ability to simulate a lower gap **coupled with** sediment transport **to evaluate** geomorphic evolution **within a numerical model**.

Ok, this is better.

L82 What is meant by a 'prototype real world location'?

For clarity we have removed the word "prototype".

Ok.

L84 A new paragraph stating a clearer and more elaborated list of aims and, or hypotheses is needed. This would help to give the paper more structure and purpose.

We have added the following text in support of this:

**The aims of this paper were to explore the relative behavioural impact of a simple LD on sediment transport processes and subsequent influences on discharge and water storage through a small reach. To do this, we first introduce new functionality for CAESAR-Lisflood that can represent LDs through the restriction of flow. Second, we evaluate the sensitivity of the model to DEM resolution, and third, assess the impact of LD roughness and gap size on geomorphology and water storage. Finally, we present the implications of numerically simulating LDs coupled with sediment transport processes to inform future modelling studies.**

This is an improvement.

L159-160 'Upwinding' and 'upwind' are strange terms to use. Consider rewording?

The upwind scheme is the correct term for calculating the downstream flow field based on cells upstream, however for clarity we have updated this to:

CL employs a first order **upwind scheme (Coulthard et al., 2013).**

It would be better if this term was clearly defined here, or other terms were used to make this clearly understandable (or does this journal assume reader knowledge of the term?).

L190 This approach of scaling n seems sensible but perhaps a caveat is needed here given that it isn't based on an empirical relationship as it stands.

We have added the below caveat immediately following the introduction of BR:

**BR captures increasing LD roughness with increasing stage to reflect increasing complexity as a greater vertical area of the channel is obstructed. Further empirical data is required to assess this assumption.**

Ok this is an improvement.

L210-214 More details on the prototype reach is needed in the paper rather than citing the thesis. The channel slope used seems quite low for a headwater stream where leaky barriers are typically used. Would it be worth testing the model over a range of slopes to see the effect? Why were different DEM resolutions tested? Certainly important but little context or purpose on this is given.

We have added more details to this section as outlined below:

The model domain was 160 m long and 100 m wide **and represents a second-order stream**. The DEM had the same average slope as a prototype site (0.01 m m$_{-1}$; Wolstenholme, 2023) **where LDs were installed in 2019** and was created by linear interpolation between the high and low survey points in the reach captured with a Topcon OS-103 Total Station (TS).

We explored the impact of grid resolution to understand whether any bias was introduced in the model outputs, and to ensure that potential caveats as a result of resolution were highlighted in this development and technical paper. We agree that it would be interested to explore the impact of slopes on the effect, however this is beyond the current scope of the manuscript.

This helps but any details to give on the topographical survey in terms of survey strategy and point density? This can affect the quality of the topography used. What catchment area is the reach at? What is the channel width? Could testing different slopes be referred to as an aspect for further study at the end of the paper?

L221 What is meant by sediment types?

Sediment types referred to the different grain size bins used in CAESAR-Lisflood, however upon reflection this was confusing. Therefore, we have adjusted the sentence as follows:

This ensured that **the sediment was** distributed throughout the catchment in equilibrium with the topography.

Ok fine.

L238 Again like testing a range of slopes, it would be interesting to see the effects of a range of different flow events and perhaps would provide more insight into hydrogeomorphic effects of leaky barriers than testing a range of different DEM cell sizes.

As mentioned above, we agree that this would be interesting, however is beyond the scope of the development and technical paper where we focus more on the implementation of leaky dams in CAESAR-Lisflood.

Ok could this be referred to as an aspect for further study?

L246 Why was an nmax value of 0.16 used?

This was chosen as a conservative estimate of the leaky dam roughness based on empirical studies (e.g., Curran and Wohl, 2003; Dixon et al., 2016 and Addy and Wilkinson, 2019). Second, we explore the relative impact of nmax in section 4.3 from 0.12–0.2, and for the purposes of not biasing the impact of grid resolution or gap size with changing multiple parameters in parallel, we chose the central value (0.16) of this range.

To provide this justification in the text, we have made the following adjustments:
...dams was set to 0.16 **(chosen as a conservative estimate of LD roughness after Curran and Wohl, 2003; Dixon et al., 2016; Addy and Wilkinson, 2019 )**.

Ok this is clearer.

Figure 6 To make it clear, mark on erosion and deposition labels. I.e. negative values show deposition and positive values show erosion which may be counter-intuitive at first glance. Similar remarks can be made for Figure 7 in relation to changes in water storage.

Negative values represent erosion and positive, deposition. We inverted the standard output from CAESAR-Lisflood to ensure that the elevation change was not misconstrued. We have, however, added text to that effect to Figure 6 to ensure that there is no confusion.
Figure 7 only shows water storage therefore we have not altered this figure.

Ok good, couldn't see the new revised figure to check this though.

L343-344 The finding that increasing LD gap size resulted in less sediment being lost seems counter-intuitive.

Less sediment was lost out of the model domain. There was less erosion upstream and downstream of the leaky dams when gap size was increased (e.g., Figure 9) as the river had less energy to erode the bed when compared to a smaller gap size.

Ok if that is the case but perhaps need to be clear that is for a condition of no sediment supply coming from upstream into the reach at the upstream boundary (if that was the case). I.e. are these geomorphic findings for a clear water scour condition? Adding a log jam in that blocks more of the channel for the latter condition would be expected to result in reduced sediment yield downstream.

L455-459 Perhaps a more important point is how valid is this model. Whilst there may be some value in using it in an heuristic manner for scenario testing, validation using empirical observations of changes in discharge and morphology would strengthen the value of this model and give practitioners more confidence in using it. This is a knowledge gap that should be stated. The word 'behaviouralist' seems a strange one to use.

We agree that to add more value the discussion would benefit from being more closely tied in with existing leaky dam literature. Therefore, to address this point we have included the following text at the end of section 5.2.:

Care must be taken when using the LD module for CL as the right results, such as elevation change and $Q$ reduction, may be overestimated when using coarser grid resolutions. **The scenarios simulated herein align well with the relative influence of LD from both field and laboratory observations including the formation of downstream pools (Lo et al., 2021; Lo et al., 2022; Muhawenimana et al., 2023), and the potential for sediment storage upstream**

**(Comiti et al., 2008). Future work should focus on calibrating and developing this tool as a flexible and rapidly deployable option for LD simulations in CL, that should currently be used heuristically to mitigate the need for calibration. The LD module for CL can therefore be best used to understand the relative impact of LDs in larger, complex catchment to identify their individual impact on FRM.**

This is better but the words **..'that should currently be used heuristically to mitigate the need for calibration'** are unclear and should probably be removed. The fact is the model is uncalibrated and unvalidated using real world data on morphological change adjacent to LDs. The potential value of the model to make predictions at the catchment scale is true but again some testing first at the reach scale is needed to underpin this first. This includes both calibration and validation.

L477-480 Yes this is a fair statement but there simply isn't usually the data to calibrate or validate sediment transport in the real world unfortunately and processes are notoriously difficult to predict for different hydrological events. In contrast, observations of channel hydraulics and hydrology are more available o accessible to enable model assessment.

We are glad that the reviewer agrees with this statement, and that the lack of data to calibrate/validate sediment transport highlights the need for the heuristic approach adopted and subsequent analysis.

Ok but model outputs in terms of erosion and deposition responses could be used for calibration and validation instead of sediment transport. Perhaps mention this earlier (see point above)?

L491 Could the model be used to assess other types of NFM structure or in channel wood? Another aspect deserving further study in future?

Absolutely. We have added the following to the previous paragraph to highlight this:

**Additionally other types of structures with porosity can be evaluated with the model, if it can be defined through its height, gap (or lack of) and roughness, for example natural wood, bridges and bunds, and presents opportunities for further study.**

Good though some of these structures are not necessarily porous (bridges, bunds).

Associated files with the paper. Appears that the model has been uploaded for both weblinks on Zenodo. Could not see the datasets.

All data used in the study and the model are available on Zenodo, in the .7z file.

Ok although the file format wasn't readable (CAESAR model needed?).

**Typo errors**

L467-468 This sentence does not make sense.

Apologies, this has been corrected to:

**Sediment transport becomes increasingly important when unintended geomorphic adjustment to 'hard engineered' structures reduces the efficacy of structures, potentially increasing flooding downstream (Hesselink et al., 2003; Pinter et al., 2006; Hudson et al., 2008; Benito and Hudson, 2010).**

Ok.

**References**

Follett, E. and Hankin, B., 2022. Investigation of effect of logjam series for varying channel and barrier physical properties using a sparse input data 1D network model. Environmental Modelling & Software, 158: 105543.

Geertsema, T.J., Torfs, P.J., Eekhout, J.P., Teuling, A.J. and Hoitink, A.J., 2020. Wood-induced backwater effects in lowland streams. River Research and Applications, 36(7): 1171-1182.

Lo, H.W., van Leeuwen, Z., Klaar, M., Woulds, C. and Smith, M., 2022. Geomorphic effects of natural flood management woody dams in upland streams. River Research and Applications, 38(10): 1787-1802.

Muhawenimana, V., Wilson, C.A., Nefjodova, J. and Cable, J., 2021. Flood attenuation hydraulics of channel-spanning leaky barriers. Journal of Hydrology, 596: 125731.

Norbury, M., Phillips, H., Macdonald, N., Brown, D., Boothroyd, R., Wilson, C., Quinn, P. and Shaw, D., 2021. Quantifying the hydrological implications of pre- and post-installation willowed engineered log jams in the Pennine Uplands, NW England. Journal of Hydrology, 603: 126855.

Schalko, I., Lageder, C., Schmocker, L., Weitbrecht, V. and Boes, R.M., 2019. Laboratory Flume Experiments on the Formation of Spanwise Large Wood Accumulations: Part II—Effect on local scour. Water Resources Research, 55(6): 4871-4885.

van Leeuwen, Z.R., Klaar, M.J., Smith, M.W. and Brown, L.E., 2024. Quantifying the natural flood management potential of leaky dams in upland catchments, Part II: Leaky dam impacts on flood peak magnitude. Journal of Hydrology, 628: 130449.